# Sea-ice evaluation of NEMO-Nordic 1.0: a NEMO–LIM3.6 based ocean–sea ice model setup for the North Sea and Baltic Sea

Per Pemberton[1], Ulrike Löptien[3], Robinson Hordoir[2], Anders Höglund[2], Semjon Schimanke[2], Lars Axell[2], and Jari Haapala[4]

[1]Swedish Meteorological and Hydrological Institute, 426 71 Gothenburg, Sweden
[2]Swedish Meteorological and Hydrological Institute, 601 76 Norrköping, Sweden
[3]GEOMAR Helmholtz Centre for Ocean Research Kiel, Düsternbrooker Weg 20, 24105 Kiel, Germany
[4]Finnish Meteorological Institute, FI-00101 Helsinki, Finland

*Correspondence to:* Per Pemberton
(per.pemberton@smhi.se)

**Abstract.** The Baltic Sea is a seasonally ice covered marginal sea in northern Europe with intense wintertime ship traffic and a sensitive ecosystem. Understanding and modeling the evolution of the sea-ice pack is important for climate effect studies and forecasting purposes. Here we present and evaluate the sea-ice component of a new NEMO–LIM3.6 based ocean–sea ice setup for the North Sea and Baltic Sea region (NEMO-Nordic). The setup includes a new depth-based fast ice parametrization for the Baltic Sea. The evaluation focuses on long-term statistics, from a 45-year long hindcast, although short-term daily performance is also briefly evaluated. We show that NEMO-Nordic is well-suited for simulating the mean sea-ice extent, concentration and thickness as compared to the best available observational dataset. The variability of the annual maximum Baltic Sea ice extent is well in line with the observations, but the 1961–2006 trend is underestimated. Capturing the correct ice thickness distribution is more challenging. Based on the simulated ice thickness distribution we estimate the undeformed and deformed ice thickness and concentration in the Baltic Sea, which compares reasonably well with observations.

## 1 Introduction

The Baltic Sea is seasonally ice covered and in the northern part the sea-ice season can last for up to 7 months. The maximum total sea-ice extent is usually reached in late February and between mid-February and mid-March the ice covers on average 45% of the total Baltic Sea area. However, interannual fluctuations around this mean are very large and during severe winters the entire Baltic Sea can be completely ice covered (Leppäranta and Myrberg, 2009; Vihma and Haapala, 2009).

With 15% of the world's cargo transportation, the Baltic Sea is one of the heaviest trafficked seas in the world (HELCOM, 2009). Despite the harsh wintertime sea-ice conditions intense maritime traffic proceeds through-out the year with ships continuously operating to the northernmost ports of the Baltic Sea. This usually requires some assistance by ice-breakers and traffic restrictions based on the ship's ice class are therefore imposed by the ice-breaking authorities (HELCOM, 2004).

Regular sea-ice forecasts are thus vital to support the intense ship traffic (Löptien and Axell, 2014) and numerical ocean–sea ice models together with satellite and ship-based observations are used by the operational Ice Services around the Baltic Sea.

Here not only the extent of the ice cover is of interest, but also ice thickness and information about the ice types are highly relevant. Close to the coasts and in shallow areas the ice usually appears as fast ice while it is drifting elsewhere (Leppäranta and Myrberg, 2009). As a consequence of surface and bottom stresses, due to winds and ocean currents, convergent ice motion can lead to deformations of the ice pack. This creates so called ice ridges, barriers of very thick ice, which can be up to 30 m thick (Leppäranta and Myrberg, 2009). In addition, a convergent ice field can lead to high ice pressure that can severely hinder the ice-going traffic. It is therefore of high importance for the maritime traffic to monitor and forecast the extent of ridged ice and ice pressure (Leppäranta and Hakala, 1992; Löptien et al., 2013).

In addition, results from sea-ice models have also been used for understanding winter maritime traffic and analyzing winter ship navigation accidents (Goerlandt et al., 2016), building a winter navigation risk model (Valdez Banda et al., 2015), and developing data-driven models for ship performance in ice (Montewka et al., 2015).

For the climate system sea ice constitutes a barrier that strongly reduces the exchange of heat, nutrients and gases as well as impacts the momentum transfer between the ocean and atmosphere. Here numerical ocean–sea ice models have been used as the main tool to understand how changes in the climate system would impact the state of the Baltic Sea (Haapala et al., 2001; Meier, 2006), how changes in the ice cover affects biogeochemistry (Eilola et al., 2013) and how changes in sea ice impacts Baltic ringed seals (Meier et al., 2004).

Clearly the need to model sea-ice processes is an integral part of the ocean–sea ice forecasting task, crucial for maritime winter traffic studies as well as an important component of the coupled ocean–ice–atmosphere climate system. Even though the physical processes at work are the same on these different time and spatial scales, model limitations have called for different models systems to be used at the Swedish Meteorological and Hydrological Institute (SMHI) in the past for ocean–sea ice forecasting (Funkquist and Eckhard Kleine, 2007) and climate studies (Meier et al., 2003) of the Baltic Sea. The development of such models is time consuming and a common model system would thus be beneficial. Here the use of a community model system such as e.g. NEMO (Nucleus for European Modelling of the Ocean) (Madec, 2016) provides an excellent tool to keep up with the state-of-the-art model development. This approach has been adopted at SMHI where a new NEMO-based configuration – NEMO-Nordic – used both for forecasting and climate purposes, has been setup for the Baltic and North seas.

The scope of this article is to present the sea-ice component of NEMO-Nordic to the community. This is done by evaluating a 45-year long hindcast simulation of the Baltic Sea with a set of observations. The article is outlined as follows: in section 2 we describe the model, simulation, metrics and observational data set that are used. In section 3 we evaluate the new configuration focusing on the 45-year long hindcast simulation, followed by a summary and conclusions in section 4. Note that the ocean component of NEMO-Nordic is presented and evaluated in Hordoir et al. (2015) and in a forthcoming separate article (Hordoir et al., A NEMO based ocean model for Baltic & North Seas, research and operational applications; in preparation).

## 2 Methods

### 2.1 Model Description

The model framework NEMO and the integrated Louvain-la-Neuve sea ice model (LIM) provides possibilities to simulate ocean and sea ice processes on a multitude of time and space scales with applications ranging from global climate simulations to regional forecasts. Here we describe our NEMO-Nordic setup that uses the stable NEMO-LIM3 version 3.6 in a regional configuration covering the North and Baltic seas. Below we briefly describe the different components of the configuration with the specific choices of parameter settings and physics options.

#### 2.1.1 The Ocean Model

The main model domain of NEMO-Nordic covers the English Channel, North Sea and Baltic Sea. In the present study we use a sub-region of the main domain covering only the Baltic Sea and Kattegat to save computational time. This setup has the same physical options, horizontal and vertical resolutions as the larger North Sea/Baltic Sea domain. The only difference is that the open boundary is in Kattegat instead of in the English Channel and the North Sea. The effect of omitting the Skagerrak and North Sea region is very limited for the Baltic Sea ice state as the sea-ice growth and melt is mainly driven by the surface fluxes rather than the advective signal. However, sea-ice can occasionally form along the Swedish west coast in the Skagerrak region and this is obviously not simulated in this setup.

NEMO-Nordic's horizontal resolution is 0.055° in the zonal and 0.033° in the meridional direction. This amounts to a nominal resolution of 3.7 km (2 nautical miles). Compared to the first baroclinic Rossby radius, which is 2–11 km in the Baltic Sea (Alenius et al., 2003; Osinski et al., 2010), this makes the model operating in an eddy-resolving to eddy-permitting regime. The vertical resolution is 3 m in the upper layers down to 60 m, and then gradually increases to 22 m at depth, with a total of 56 layers. The vertical discretization uses the $z^*$ formulation and the bottom topography is represented by the partial steps approach. The setup utilizes a fully non-linear free surface formulation with a time splitting of barotropic and baroclinic modes to speed up simulation time. The ocean model time step is 360 seconds and the ice model is called every 5th time step. Vertical mixing is represented by the two equation generic length scale formulation (Umlauf and Burchard, 2005). In addition, Laplacian horizontal and isopycnal mixing is used in conjunction with a bottom-boundary layer parametrization (Beckmann and Döscher, 1997).

For further details and evaluation of the NEMO-Nordic ocean model setup the reader is referred to Hordoir et al. (2015).

#### 2.1.2 The Sea Ice Model

LIM3 is a dynamic–thermodynamic sea-ice model with a multi-category ice thickness distribution and multi-layer halo-thermodynamics (Vancoppenolle et al., 2009; Rousset et al., 2015). The ice dynamics use a modified elastic-viscous-plastic (EVP) rheology (e.g. Bouillon et al., 2009) and accounts for sea-ice deformation processes (ridging and rafting).

The present NEMO-Nordic builds upon version 3.6 of LIM3. Compared to the polar oceans, sea ice[1] in the Baltic Sea is only seasonal, generally thinner, and with a much lower brine content due to the low salinities in the Baltic Sea. Thus, some model parameters need to be adjusted to the Baltic Sea conditions. In Table 1 we show the settings of the physical sea ice parameters that were adjusted in our NEMO-Nordic setup. The settings are also compared to that of a large-scale global ocean

configuration ORCA2-LIM3, a configuration that is included in the NEMO–LIM3.6 model system. Below we briefly describe the rationale behind most of these settings.

In NEMO-Nordic the ice thickness distribution is discretized using 5 different categories and the thermodynamic calculations use 2 vertical layers of ice and 1 layer of snow. When new ice forms in open water it is assumed to have a thickness of 0.01 m. In our setup we neglect all internal halodynamical processes of the sea ice, as initial tests using this option yielded an

unstable model, particularly close to river mouths. Instead we use a constant bulk salinity of 0.001 g kg$^{-1}$, which essentially means that the effect of brine pockets is neglected. This value is 10% of the river salinity used in the model and was chosen for numerical stability reasons. We tested an ice salinity of 0.0 g kg$^{-1}$ but that yielded an unstable model, likewise with an ice salinity higher than the river water salinity. For numerical stability reasons ice models also require some horizontal diffusion. In our configuration, with a relatively high horizontal resolution, we use a relatively low diffusivity constant of 1.0 m$^2$ s$^{-1}$.

LIM3 has a ridging scheme that accounts for the thickness growth due to sea-ice ridging. In this scheme there is a parameter rn_hstar that adjusts the upper bound of the ridged ice thickness. Since ridges are generally thinner in the Baltic Sea compared to polar oceans we lowered rn_hstar to 30.0 m from the default 100.0 m, likewise we lowered the crossover thickness for when sea ice ridge instead of raft from 0.75 m to 0.07 m, with a more sharp transition. This value is a Baltic Sea adaption of the analytical modeling work by Parmerter (1975), who suggest 0.17 m for Arctic conditions. The ice and snow albedos use the

default formulation in LIM3 and the snow conductivity the default value of 0.31 W m$^{-1}$ K$^{-1}$.

In addition, we have implemented a simple fast ice parametrization since this is not included in the present version of LIM3. Fast ice is an important feature of the Baltic Sea ice cover and usually occurs in coastal and archipelago regions where the ocean depth is shallow. This is done by simply masking out grid points where the depth is below 15 m in the dynamical components of LIM3 so that the ice remains stationary. The fast ice mask is deactivated if the total ice volume in a cell is below 0.001 m$^3$,

which means that for extremely low concentrations there can be advection of ice in the fast ice zone. The main effect of the fast ice parametrization is that a region of more or less undeformed ice is formed close to the coasts, and that the ridges are formed outside of this region.

## 2.2  Ice thickness distribution

On spatial scales of $O$(km), the scale of the present sea-ice model, the ice thickness varies considerably due to both thermo-

dynamical growth and mechanical redistribution of ice. To account for such sub-grid scale ice thickness variations sea ice is described in terms of an ice thickness distribution $g(h)$, following Thorndike et al. (1975). Here $g(h)dh$ gives the areal fraction

---

[1]In fact, on a micro-scale it is brackish ice rather than sea ice. However, the brackish Baltic Sea ice can still be assumed to behave as sea ice (Leppäranta and Myrberg, 2009).

of ice with a thickness between $h$ and $h + dh$. From the distribution we can calculate the ice concentration $A$ as

$$A(x,y,t) = \int_0^\infty g(h,x,y,t)\,dh, \tag{1}$$

and the mean thickness as

$$H(x,y,t) = \int_0^\infty h\,g(h,x,y,t)\,dh. \tag{2}$$

In LIM3 the ice thickness distribution is discretized by defining $n$ ice categories with thickness bounds $H_i^{lower}$ and $H_i^{upper}$ for category $i$. Usually 5 categories are sufficient to resolve the sub-grid scale distribution (Bitz et al., 2001). Within each category the ice thickness is free to evolve between $H_i^{lower}$ and $H_i^{upper}$, and as thermodynamical and dynamical processes form or melt the ice, the ice thickness changes and LIM3 accordingly remaps the ice thickness distribution to account for this.

For regional applications, where sea ice usually is thinner than in the polar regions, LIM3 has a new scheme to calculate the
ice category bounds (Rousset et al., 2015). Based on an expected domain average ice thickness $\overline{h}$ the category bounds are fitted to a function $(1 - h)^\alpha$ on the interval between 0 and $3\overline{h}$. In NEMO-Nordic the ice thickness distribution is discretized using 5 different categories based on a $\overline{h} = 0.5$ m giving the lower bounds: 0.0, 0.25, 0.56, 0.95 and 1.46 m (also shown in Fig 10a).

To compare the simulated thicknesses with observations we use two different metrics:

$$\overline{H} = \frac{\sum_{i=1}^5 g_i h_i}{\sum_{i=1}^5 g_i}, \tag{3}$$

which is the mean ice thickness for each grid cell, the discrete counter-part of Eq. 2 (also called cell-averaged thickness); and

$$\overline{H}_{level} = \frac{\sum_{i=1}^4 g_i h_i}{\sum_{i=1}^4 g_i}, \tag{4}$$

which is a proxy for the undeformed level ice. The upper bound for the fourth category is 1.46 m which is greater than the maximum thermodynamical ice growth for most conditions of the present Baltic Sea state. In addition, there is a distinct separation in the ice thickness distribution between the first four and the last category, see Fig. 10 (this is discussed more in
section 3.3). We interpreted this as representing a separation between the thermodynamically and dynamically grown ice, and thus use the first four ice categories as a proxy for level ice and the fifth category as a proxy for ridged ice. We stress that this is just an approximation as rafting and smaller ridges will also be represented by the model in the lower categories. In an effort to assess the precision of this approximation we ran a 5 year test where we turned of all mechanical deformation of ice, and then compared it to a control simulation for the same period to get an estimate of thermodynamically grown ice in the fifth
class. This resulted in ice volumes (in category five) of 6–20% in the case with no deformation, however, the ice in the lower ice classes was also strongly impacted due to the missing transfer of ice, and the test was deemed too artificial to assess the precision of the approximation. However, for many applications (e. g. maritime winter traffic) it is the actual thickness rather than the underlying processes that is important.

## 2.3 Forcing and simulation

As atmospheric forcing we use downscaled ERA-40 reanalysis data (Uppala et al., 2006) which, compared to the original ERA-40 reanalysis, features additional regional details which considerably affect the solution of standalone ocean models of the Baltic Sea (Meier et al., 2011). Note that the ERA-40 data set only covers the period up until 2002 and afterwards we use operational analysis from the ECMWF (European Centre for Medium-Range Weather Forecasts) for the downscaling. Using a different data set for the last 4 years could potentially impact our results, however, our analysis shows no evidence of that. The downscaling procedure takes ERA-40 reanalysis data as boundary conditions for the regional Rossby Centre Atmosphere model (hereafter RCA) which features an enhanced (relative to ERA-40) horizontal resolution of 25 km (Jones et al., 2004; Samuelsson et al., 2010). As shown by Samuelsson et al. (2010) the approach provides a very realistic climatology. This downscaling approach was successfully used in earlier studies (e.g. Dietze et al. (2014), Hordoir and Meier (2011), Hordoir et al. (2013), Löptien and Meier (2011), Löptien et al. (2013)). The present forcing is an advancement as it uses the updated atmospheric model RCA4 and spectral nudging (Berg et al., 2013), which ensures that the simulated cyclone paths match the actual tracks.

This atmospheric forcing is applied to NEMO-Nordic in a 45-year long (1961–2006) hindcast simulation. The simulation is initialized from rest with climatological salinity and temperature distributions. The simulation starts in January 1961 with no ice present in the model. As the seasonal sea ice disappears every summer the spinup of the ice cover is usually short and already at the next winter the ice cover is well adapted. To account for the ice spinup we start our analysis in the beginning of the 1961/62 ice season. On the open boundary in Kattegat the model is forced by sea level variations from daily tide gauge data. For temperature and salinity on the open boundary and runoff draining into the Baltic Sea we use monthly climatological values. Meier et al. (2012) showed that, for temperature such an approach is sufficient for the Baltic Sea as most of the trend comes from the atmospheric forcing. However, in Kattegat close to the open boundary in the sub-region, the solution can of course be affected by the simplified boundary conditions.

## 2.4 Observational Data

We use several observational data sets to evaluate the sea ice model. An extensive historical data set, named BASIS, covers the winters 1960/61 to 78/79. This data set contains the, at that time, best available information on the ice concentration, thickness as well as on the dominant ice types. BASIS is based on hand drawn sea ice charts which were provided by the local weather services for shipping. The sea ice charts were derived from direct ice measurements and estimates from voluntarily observing ships, coast guards, ice breakers, light houses and harbour authorities. Additional information came from over-flights by FMI, SMHI and the Swedish Air Force. From the late 1960s onwards satellite observations were partly included. Thus, the underlying ice charts were extrapolated from the irregular (as regards space and time) observations. The associated uncertainties are unclear and are presumably largest away from the major shipping lines. Nevertheless, BASIS is the best available information on historic ice conditions in the Baltic Sea. These ice charts were collected and then digitized 1981 in a joint project of the Finnish Institute of Marine Research (today FMI) and SMHI. The original data were hard to access as

BASIS-ice was designed for storage on punchcards. Thus, Löptien and Dietze (2014) provided an easier to access version in the free file format NetCDF via www.baltic-ocean.org (or PANGEA doi:10.1594/PANGAEA.832353). BASIS ice thicknesses was originally indexed by numbers from 1–9. These numbers were assigned to thickness classes (1–2, 3–6, 7–12, 13–20, 21–30, 31–42, 43–56, 57–72 and more than 73 cm). Thus a lower bound for the uncertainty when it comes to ice thickness is the precision given by these classes. Note that when we calculate integrated metrics, e.g. total ice extent and total ice volume, the BASIS data set is first interpolated to the same grid as NEMO-Nordic and then masked using the land/ocean mask of the ocean model. This is done to mask out Skagerrak and to have comparable total areas for the Baltic Sea.

In addition to BASIS, we use modern ice charts, called IceMaps, (which interpolate similar observations as in BASIS); the Swedish Ice Service of SMHI also provided weekly ice thickness measurements (1971–2010) in the fast ice zone at the stations Järnäs (19.41E 63.26N) and Kemi (24.31E 65.4N). These are located in the Bothnian Bay and Bothnian Sea, respectively. Additionally, we use airborne EM-ice thickness measurements in the basin interiors of the Gulf of Finland and Bothnian Bay, which were collected 23th of February 2003, 14th of March 2004 and 13th of March 2005, all within the IRIS-project (Haas, 2004). It is, however, important to keep in mind that comparing point measurements to the model accounts for very different scales and has to be considered with some caution.

To evaluate the sea surface temperature (SST) we use CTD casts from SMHI and a satellite derived data set from the Bundesamt für Seeschifffahrt und Hydrographie (BSH) (Loewe, 1996). The CTD casts where done on a close to monthly basis at the stations Anholt E, Fladen, BY15, BY31, MS4, NB1 and F9, for the location see Fig. 1 and for the time period Fig. 5. The BSH data consist of high-quality satellite SST data product compiled into a monthly data set covering the period 1990–2006. Also here it is important to be cautious when interpreting model biases as the modeled SST represent the temperature of the upper 3 m whereas the CTD cast and BSH represent the near-surface and surface values, respectively. For the snow thickness evaluation we use data of annual maximum thickness from two different stations, Kemi and Hailuoto (see Fig. 1), covering the periods 1961–2005 and 1974–2006, respectively. These data were provided by FMI. For the air temperature evaluation we use data from SMHIs meteorological archive. Here a set of observations from caisson lighthouses, lightships and small islands in the Baltic Sea where used, see map and legend in Fig. 1 for the locations and sampling periods.

## 3 Model Evaluation

In this section we evaluate NEMO-Nordic's performance against a set of different observational data sets. The main focus is on the long-term statistics of important sea-ice parameters such as sea-ice concentration, extent and thickness. For the climate system any changes in these sea-ice parameters are crucial and are thus important to evaluate for future climate studies and related studies on e.g. winter navigation and hazard. We also briefly compare single days when the ice cover reached its maximum extent, for 2 extreme winters. This is done to evaluate the model's capability to capture extremes on daily time scales which is important for forecasting purposes. However, we stress that the model data is from a hindcast simulation, which is a totally different mode of operation compared to how sea-ice forecasts are run. Before evaluating the sea-ice parameters we evaluate the simulated SST, 2 m air temperature and snow forcing.

## 3.1 Evaluation of 2 meter air temperature, snow thickness and sea surface temperature

In this section we aim at evaluating the biases of some of the forcing parameters that impact the ice cover: the 2 m air temperature, snow thickness and sea surface temperature. The parameters are chosen based on the available observations that we could gather in the seasonally ice covered model domain. We stress that to robustly assess the thermal exchanges occurring between the ice–ocean–atmosphere interfaces we lack observations of important components of the full radiative and turbulent heat fluxes, and the analysis is thus somewhat incomplete. A more rigorous evaluation of the atmospheric biases is desirable and urgent, but is beyond the scope of this article. In addition, the approach we use by forcing NEMO-Nordic with a passively coupled downscaling of ERA-40 also has its shortcomings e.g damping of heat anomalies and biases related to the prescribed lower boundary conditions of the atmospheric model. Some of these issues are investigated and discussed by Gröger et al. (2015).

We now compare observed 2 m air temperatures at a selected number of locations in the northern Baltic Sea with the downscaled ERA-40 atmospheric forcing data. Figure 2 shows the long-term wintertime 2 m air temperature biases at the different stations for two periods, before and after 1979. Note that the sampling at the stations covers different time periods, see the legend in Fig. 1, and are thus not strictly comparable. The time period division of before and after 1979 is intended to reflect systematic biases for the BASIS period (the first part) and for the latter decades (the period when we have satellite observations of SST). For the first period most stations show an overestimation (downscaled ERA-40 forcing being warmer than the observations) during January–March, with Sydostbrotten in the Bothnian Bay, Grundkallen in southern Bothnian Sea and Svenska Högarna i the northern Baltic Proper standing out most with maximum biases around 0.5–2.0 $^{\circ}C$. In April, on the other hand, some stations show a negative bias while other show a positive bias with a range of -0.8–0.9$^{\circ}C$. Here Finngrundets Fyrskepp, Grundkallen and Svenska Högarna suggest a positive bias in the southern Bothnian Sea/northern Baltic Proper. The Sydostbrotten station yields a negative bias in the Bothnian Bay for April, same for the Västra Banken Aut and Gustaf Dalén Aut, however the latter two only covers 3 years for this time period and might thus not be representative. For the second period some stations (these are only located in the Bothnian Bay) show a large positive (1.1–2.2 $^{\circ}C$) bias for January–March, while other stations show a more mixed signal with either a smaller negative or positive bias in the range -0.5–0.7$^{\circ}C$. It is hard to distinguish a strong coherent geographical signal for the January–March period as nearby stations covering the same period can give incoherent biases, e.g. the January bias at Rödkallen and Pite-Rönnskär showing both positive and negative offsets respectively. In April on the other hand, the signal is more coherent with many stations both in the Bothnian Bay Bothnian Sea and northern Baltic Proper showing negative biases of -1.0 to -0.5$^{\circ}C$. We caution that the observations are made on different measurement platforms which might have a local climate depending on the location. For instance, observations made on small islands (see legend in Fig. 2 for the measurement platform type) might be impacted by land, which due to the resolution is not present in the downscaling, and might therefore show larger seasonal cycles and are thus not representative of the temperature over ocean. Here e.g. Pite-Rönnskär and Holmön stand out compared to other stations. On the smaller platforms (caisson lighthouses and lightships), on the other hand, the representability of the over ocean temperature can be assumed to be better.

We now continue to evaluate NEMO-Nordic's snow thickness. Due to its relatively low thermal conductivity the snow pack has an isolating effect on the underlying sea ice, and any biases in the simulated snow thickness will impact the ice growth and melt. Figure 3 shows a comparison of the simulated and observed maximum annual snow thickness from two station in the Bothnian Bay. For these two stations it is clearly seen that the model has a problem simulating the observed snow cover. For Hailuoto years with a thinner snow cover (10–25 cm) the simulated snow cover has a smaller offset, less than 3 cm, while for years with a thicker snow cover there is a much larger offset with the model underestimating the snow cover by 8–37 cm. At Kemi there is an overestimation by up to 11 cm for the years with a thinner snow cover, while for years with a thicker snow cover the model has, similar to Hailuoto, a strong underestimation of 7–37 cm. The temporal correlation at Kemi is quite low (0.35), while at Hailuoto it is somewhat higher (0.60). By inspecting the long-term change in snow thickness bias (not shown) we found the the offset is large in the beginning of the simulation and decreases towards the end of the simulation at both stations. We caution that comparing the snow thickness at only two stations is problematic due to the large spatial and temporal variability of the snow cover. However, our results with an underestimation of the snow cover around the Bothnian Bay and Bothnian Sea are also in line with what Samuelsson et al. (2010) found for a number of stations along the Swedish coast in this region, in their evaluation of the RCA model.

The SSTs reflect the air-sea interaction of heat and biases in the SSTs indicate that either the atmospheric forcing and/or the ocean dynamics could be misrepresented. Any such biases will also affect the growth of sea ice and we therefore briefly evaluate the SST biases in NEMO-Nordic in the following. Here we compare the simulated SST with a satellite derived SST product from BSH and with CTD casts at a few stations in the Baltic Sea and Kattegat. Figure 4 shows the long-term (1990–2006) wintertime SST biases over the model domain. We caution that in regions which usually are ice-covered these means are heavily weighted to ice-free conditions as the satellite sensor has limited capabilities to estimate under-ice SSTs. We have therefore masked out these areas using an observed ice concentration greater than 50 % as an indication of suspicious points. As seen in Fig. 4 NEMO-Nordic tends to be colder than the BSH data set over most of the model domain, for the January–March period. An exception is in the Gulf of Riga where there is a persistent positive bias of $\sim 0.5^\circ C$. Some parts of the central Baltic Proper and Gulf of Finland also exhibit a small positive bias (less than 0.1 $^\circ C$). In April, on the other hand, NEMO-Nordic has a pronounced positive bias (of up to $0.8^\circ C$) in the coastal regions of the Baltic Proper, Bothnian Sea and the outer Gulf of Finland. In the central parts of the Baltic Proper the bias is very small (less than $\pm 0.2$). Calculating the area mean bias over the Baltic Proper yield a change from a negative $-0.5$ bias for January–March to a small positive bias (less than $0.1^\circ C$) for April. The Bothnian Sea and Gulf of Finland, on the other hand, all have negative area mean biases for all winter months. Another evident feature is that there is a persistent strong negative bias in Kattegat, particularly pronounced in January and February. The January–April mean bias for the entire Kattegat region is $-1.2^\circ C$.

The BSH data only cover the last two decades of the simulation and to further evaluate the simulated SSTs we now compare NEMO-Nordic with CTD data from seven different stations which have long-term monitoring, three in the Bothnian Bay and Bothnian Sea (F9, NB1, MS4), two in the Baltic Sea (BY15 and BY31) and two in Kattegat (Fladen and Anholt E). All available data in the upper 3 m is averaged for each date. Note that the depth sampling can change from station to station. Similarly to the air temperature bias analysis we have divided the data into two periods, before and after 1979, see Figure 5. For the first period

only data in the Baltic Sea and Kattegat are available. As seen the station representing the central Baltic Proper (BY15) has a positive bias with a maximum offset in April of 1.1 $°C$. The station representing the western Baltic Proper (BY31) has negative bias in January, and a positive bias in March and April. In Kattegat the Fladen station shows a large negative bias (-1.0 $°C$) in January, that is reduced in February and shifted towards a positive bias in March and April. For the second period there is also

data from the Bothnian Bay and Bothnian Sea. This period also covers the satellite data period. Here almost all stations (except BY15) have negative biases for the January–March period, with the largest biases seen in January, in Kattegat $\sim -1.4°C$. In April, the Bothnian Bay, Bothnian Sea and Kattegat stations have negative biases while the Baltic Proper stations have positive biases. Clearly the systematic biases seen in the stations agree quite well on the sign and magnitude of the offset compared to the satellite derived data.

To end this section we summarize all the biases seen in the 2 m air temperature, snow thickness and SSTs. For snow thickness we see a strong underestimation, especially for years with a thicker snow cover and the variability at the investigated stations is quite poor both in timing and amplitude. From the SST biases we see a too cold Kattegat region, particularly in January and February. This seems to persist throughout most of the simulation. These results are inline with the study by Gröger et al. (2015) who found that the dynamical downscaling of ERA-40 has a cold air temperature bias, due to a cold bias in the prescribed SST.

This region is also close to the open boundary, and the relatively simple boundary conditions could also impact the SSTs in this region. For the Baltic Sea the signal is different. During the latter period we see a cold bias for most of the Baltic Sea, January–March. In April on the other hand, the northern parts are still too cold, while the central parts are too warm. The too warm SSTs in April are also present for the first period. The 2 m air temperature biases give a much more incoherent picture with biases in opposite directions at nearby stations. We note that in Samuelsson et al. (2010) they found that land areas in

northern Scandinavia experience a positive 2m air termperature, while southern Sweden showed a negative bias. Comparing the 2 m air temperature biases to the SST biases is challenging, at some nearby stations (e.g. Gustaf Dalén Aut and BY31) the sign of the offset is in the same direction, while at others it is in the opposite direction (e.g. Landsort A and BY31). As we have no information on the biases in the shortwave, long wave and latent heat fluxes it is hard to attribute the causes of the two main signals: the basin-wide cold SSTs, and the warm central Baltic Proper in April. We can speculate that it is related

to the prescribed lower boundary conditions (SSTs and ice cover) which impacts the cloud cover and radiative fluxes in the downscaling (e.g. Gröger et al., 2015; Hunke and Holland, 2007). This in turn impacts the onset and retreat of the ice cover. It is, however, beyond the scope of the present study to further explore the underlying reasons behind the SST biases.

## 3.2   Sea-ice concentration and extent

The extent of the sea-ice cover and its concentration are two important parameters that a sea-ice model needs to simulate well

both from a climatological and forecasting perspective. Here we show the long-term spatial coverages and the time variability of the total sea-ice extent in the Baltic Sea.

    Figures 6 and 7 show the long-term (1961–1979) monthly mean sea-ice concentrations for both NEMO-Nordic and the BASIS data set. As seen the general agreement is quite good for this period. In January both the model and BASIS agree on the coverage in the Bothnian Bay, Bothnian Sea, Gulf of Finland and Gulf of Riga. However, the ice coverage in Kattegat is

much overestimated in NEMO-Nordic. In February when the Baltic Sea ice coverage usually reaches its maximum NEMO-Nordic and BASIS agrees very well showing that the Bothnian Bay and Sea, Gulf of Finland and Gulf of Riga are completely ice-covered. The Belt Sea and Kattegat are also completely ice-covered but with much lower ice concentration while only the coastal regions in the Baltic Proper and Arkona Sea on average are ice-covered. In March the agreement is also good although

NEMO-Nordic simulates some sea ice in Kattegat. In April the reduction of sea ice is evidently stronger in the model which shows much less ice in the Bothnian Sea and Gulf of Riga. We note that the too large ice cover in Kattegat, in January (and February), and the too low ice cover in the central perts of the Baltic Proper is consistent with the evaluated SST biases at Fladen and BY15, respectively.

The annual maximum Baltic Sea ice extent (MBI) is a widely used metric to describe climate variability in the region, and

the first recordings date back to 1720 (e.g. Vihma and Haapala, 2009, and reference therein). To evaluate the MBI we compare NEMO-Nordic with observational estimates from BASIS extended with IceMaps from the Swedish Ice Service. Note that in this study sea-ice extent is calculated as the area where sea-ice concentration is at least 15%; and that we have excluded sea ice in the Skagerrak region in the observational estimates as this region is missing in the present configuration. Figure 8 shows the interannual variability of the MBI. The long-term (1961–2006) simulated and observed means (standard deviations) are

194 (78) and 167 (77) $10^3$km$^2$, respectively, and the correlation between the model and observations is 0.93. The simulated trend (-10·$10^3$ km$^2$/decade) for the 1961–2006 period is much lower than the observed trend (-23·$10^3$ km$^2$/decade). For a level of $p \leq 0.05$ the null hypothesis of no trend could not be rejected for the simulated trend ($p = 0.26$); while the observed trend ($p = 0.009$) falls well below this level. Clearly there is a shift with a change in the systematic offset occurring somewhere around the period 1973–1976, where the model goes from a period (1961–1976) with a smaller negative bias towards a period

(1977–2006) with a larger positive bias. Calculating the decadal biases yields off sets of $-1$, $-3$, 15, 20 and 28 % for the five decades. After the bias shifts to becoming positive the trends also agrees better with $-32$ and $-27$ $10^3$ km$^2$/decade for the NEMO-Nordic and observations, respectively. The SST biases at BY15 and Fladen also reflect this shift, with a transition towards more negative bias or a lower positive bias for the latter decades.

In the Baltic Sea ice winters are usually classified, following Seinä and Paluso (1996), as: mild, average, severe and extremely

severe based on the MBI. For the period 1961–2006 NEMO-NORDIC (observational estimates) shows 8 (15) mild, 25 (23) average, 10 (7) severe and 2 (0) extremely severe winters. Note that since we have excluded Skagerrak in our integration our the classification statistic are not directly comparable to other studies, and that the observations tends to show statistics weighted towards the milder end of the scale due to the regridding and masking of the data. The classification statistics and Fig. 8 show that the model underestimates the total ice extent and number of mild winters, especially during the later decades.

By inspecting year by year MBI distribution (see animation in supplemental material) we identify particularly two problem areas, Kattegat and the Bothnian Sea, where the model tend to overestimate the MBI ice cover. This occurs both for severe, average and mild winters, but impacts the observation/model intercomparison most for mild winters which are more dominant during the last two decades.

Figure 9 shows the long-term mean monthly seasonal cycles and standard deviations. It is seen that there is a faster increase

in ice cover growth and a slightly earlier retreat of the ice cover on average in NEMO-Nordic. The simulated monthly annual

maximum of 147 $10^3$km$^2$ is 18% higher than the observed ice cover and occurs in February rather than March, based on monthly averages. The maximum monthly simulated standard deviation of 69 $10^3$km$^2$ is 3% higher than the observed standard deviation, occurring the same month (February). By separating the first two decades and from the last three decades (not shown) we see that the seasonal cycle in NEMO-Nordic is well in line with the observational estimates for the first two decades, with

better a match of the seasonal cycle and an error of less than 9% in the monthly based annual maximum, while for the latter period the seasonal cycle is overestimated by 29%.

Based on daily data we now discuss the day of MBI. For the full period (1961–2006) we see that on average the MBI occurs on the 53rd and 56th day of the year for NEMO-Nordic and the BASIS/IceMap data set, respectively, i.e. in the end of February. Some years it can occur early in January, both in the model and observations, while for most years it tends to

occur in the period mid February to mid March. Inspecting the absolute difference in the day when the MBI occurs reveals an average offset of 9 days between the model and the observations. Some years the difference is quite extreme with up to 50 days offset, while most years it occurs within a week of the observed maximum. As the BASIS/IceMap data mostly is produced on an 1–2 times per week basis a better precision than within a week can not be expected. For years when the difference is large, the winter conditions usually experience two or more cold spells which could be separated by up to a month, and the simulated

maximum then occurs during a different cold spell compared to the observed maximum.

Our analysis shows that overall NEMO-Nordic agrees reasonably well with the BASIS data sets in terms of sea-ice concentration and extent, and variability of the MBI. We identified problem areas in Kattegat and the central Baltic Sea/Bothnian Sea which leads to a too high sea-ice extent during the MBI. This leads to an overestimation of the total sea-ice extent particularly the in last three decades. From the long-term seasonal cycle of total sea-ice extent we see a faster growth and a too early ice

retreat in spring. Most of these offsets in the ice cover are in accordance with the already discussed SST biases, however, we lack information on the observed heat fluxes at the ice–ocean–atmosphere interfaces preventing us to attribute the driving causes of these anomalies.

### 3.3  Sea-ice thickness, volume and deformation

Sea-ice thickness is another important sea-ice parameter that both reflects the thermodynamical and dynamical evolution of the

ice pack. Here we evaluate the sea-ice thickness distribution as well as the long-term mean sea-ice thickness, sea-ice volume and ice concentration in the thickest category, where the latter is used as a proxy for the sea-ice ridge concentration.

The sea-ice thickness distribution yields information of both the thermodynamical and dynamical ice growth. The dynamical ice growth tends to affect the extreme ends of the distribution when ridging and rafting creates thicker ice and open water while thermodynamical processes tend to populate the distributions towards a center value (Weeks, 2010). Figure 10 shows the

simulated and observationally estimated sea-ice thickness distributions, calculated as an average of the three days during 2003–2005 winters when the EM-ice thickness measurements were carried out. The EM-bird system measures the total thickness of sea-ice and snow thickness, and we thus include both these quantities in the NEMO-Nordic distribution. As seen in Fig. 10b the EM data sampling is quite sparse and mostly reflects the conditions in the central Bothnian Bay and Gulf of Finland, regions where we expect to find a wide range of thicknesses, including ridges. The mean thicknesses of NEMO-Nordic and the

observational estimates are 84 and 113 cm, respectively. Our snow thickness analysis indicated that the model has a too thin snow cover. This could perhaps be a contributing factor to the lower mean thickness, however, it is unlikely that it explains the rather striking difference in the shape of the distributions. Compared to the observational estimates NEMO-Nordic has a more bi-modal distribution which overestimates the thickness in the ranges $0.4 - 0.5$ m and $3.0 - 4.0$ m while sea-ice in the

thickness range $1.2 - 2.8$ m is strongly underestimated. Capturing the correct ice thickness distribution is clearly challenging and several things probably affect the discrepancies. (i) the observational data set is sampled on a finer spatial scale, and thus resolves a larger thickness range before it is averaged onto the NEMO-Nordic grid. (ii) the choice of ice category bounds can impact the thicknesses range. However, sensitivity tests with 10 categories and two different settings of the expected mean ice thickness (rn_hicemean=0.5 or 1.5 m) essentially yielded in the same distribution as in Fig 10. (iii) the choice of maximum

ridge ice thickness limit (rn_hstar) in the ridging parametrization sets the upper limit of possible ice thicknesses, and lowering it would shift the ridging peak towards the low end of the spectra. Another possibility to improve the ice thickness distribution, that we did not explore, would be to change the transfer functions in the ridging scheme. We also note that the model used in Löptien et al. (2013) that utilizes adaptable ice category limits seems to better resolve the Baltic Sea ice thickness distribution, particularly in the $1.2 - 2.8$ m range (c.f their Fig. 7a with our Fig. 10a). However, Fig. 10a also shows that there is a distinct

separation between dynamically and thermodynamically grown ice in the model just below the lower limit (1.46 m) of the thickest ice category. This gives merit to using the four lowest ice categories as a proxy for level ice and the thickest ice category as a proxy for ridged ice. Hence we apply this concept as NEMO-Nordic does not have an explicit ice category for ridged ice.

    We now compare the proxy level ice thickness, proxy ridged ice concentration and the area-integrated ice volumes in NEMO-

Nordic with observational estimates from BASIS. The level ice thickness in the model is calculated as a category-weighted average of the first four ice categories, see Eq. 4. This metric mainly reflects the thermodynamical growth of the model. As seen in Figure 11 NEMO-Nordic and BASIS agree very well on both the absolute magnitude and the spatial gradients in the level ice thickness. Both show the transition towards gradually thicker ice in the northern Bothnian Bay and eastern Gulf of Finland, there is also a gradient with thicker ice close to the coasts and thinner ice more central in the basins in both sources. The main

difference is an overestimation by the model of the ice thickness in a thin band in the fast ice zone, close to the Finnish coast, for both months. From the long-term January–April level ice thickness means we then calculate the area-averaged level ice thickness for the total Baltic Sea, Bothnian Bay, Bothnian Sea and Gulf of Finland which yield for NEMO-Nordic (BASIS): 21 (18), 37 (36), 14 (11) and 22 (21) cm, respectively. This shows that there is a level ice thickness bias of 17% for the total area and a 3–27% bias for the sub-areas. Given the uncertainties of the BASIS data set the agreement must be considered to be

good.

    Figure 12 compares the proxy ridge ice concentration from NEMO-Nordic with observational estimates from BASIS. Here the agreement is generally good between the model and the observational estimates, both show a high concentration in the central Bothnian Bay, and lower concentrations close to the coast in the Bothnian Sea and eastern part of Gulf of Finland, regions where we expected to find ridges. It is also seen (see inset in Fig. 12) how the fast ice parametrization leads to much

lower deformed ice concentrations close to the coasts. However, we stress that these are two different measures of the ridging:

the observations estimate the number of ridges per kilometer while the model proxy gives the areal fraction of all ice thicker than 1.46 m in a model grid cell.

To further explore the composition of the different ice categories we calculate the area-integrated ice volume per ice category, integrated over the Bothnian Bay and the entire Baltic Sea, as an average of the January–April 1961–1979 period. From Fig. 13 we can see that a large portion (25–50%) of the ice volume is found in the thickest ice category, both for the Bothnian Bay and the entire Baltic Sea, especially late in the ice season. It is also clearly seen that there is a redistribution of ice towards thicker categories as the ice season evolves, with ∼50% of the volume in the thickest category in April. Compared to the BASIS data set the total ice volume in NEMO-Nordic is much higher (68–123%), particularly in the Bothnian Bay. Compared to the ice volume in only the first four categories, our proxy level ice volume, the match is better (1–68%). During January and February the simulated level ice volume is too high, while the March (and also April for the Bothnian Bay) level ice volume almost perfectly match the BASIS ice volume. In April the level ice volume for the entire Baltic Sea is lower than BASIS data set indicating an early ice retreat for this ice type, during this period, consistent with the early total ice extent retreat. Given the uncertainties of the ice thickness measurements in BASIS and that the observations of ridges are usually quite sparse we assume that the BASIS data mainly reflect the level ice thickness and volume. As the simulated volumes imply that the deformed ice constitutes a considerable amount of the total volume this primarily explains the differences in total ice volumes both for the Bothnian Bay and the total domain.

Figures 11–13 show that, compared to BASIS, NEMO-Nordic generally agrees well on the spatial patterns and area-averaged level ice thickness, and that, if we exclude the volume in the fifth category we have an estimate of the simulated level ice volume agreeing rather well with the observed total volume. The offset in the seasonality seen in the ice extent, is also seen in the level ice volume with a tendency to overestimate the ice production in the growth phase and ice reduction in the melt phase of the season. The too low snow thickness in the model suggest that the ice thickness and volume should be overestimated. in contrast, the warmer than observed 2 m air temperatures in the downscaling, as shown from some of our stations in the northern Baltic Sea and by Samuelsson et al. (2010) for nearby land areas, suggest that there should be an underestimation. We can speculate that the two effects counteract each other to some extent, however, also here we fail to fully attribute the cause since we miss observations of all heat fluxes at the ice–ocean–atmosphere interfaces.

## 3.4 Coastal stations in fast ice zone

The sea ice in the coastal regions around the Baltic Sea usually consists of (land) fast ice which is immobile. These areas are important platforms for both human activity and marine wild life. Here the ice pack is dominated by thermodynamical growth driven by the changes in surface air temperature. In the initial stages the wind conditions also affect the formation and extent of the fast ice. The fast ice zone also affects the atmosphere–ocean momentum interaction as the wind stress is damped out. To evaluate the model performance we first compare simulated cell-averaged ice thickness with long-term weekly ice thickness observations at two coastal stations, outside Ratan and Järnäs (see Fig. 1 for the locations). Then we proceed with comparing simulated cell-averaged ice thickness outside Luleå and Kemi with ice growth estimates based on a FDD model.

Figure 14 shows the long-term mean and standard deviations of simulated and observed sea-ice thickness on the edge of the model's fast ice zone at the Ratan and Järnäs sites. At the Ratan site NEMO-Nordic simulates a too thin mean ice cover during the growth phase of the ice season, up until the observed melt starts, implying a too slow thermodynamical growth rate. The variability for the growth phase, on the other hand, matches the observations quite well. During the observed melt phase, the simulated ice cover generally still continues to increase for 1–3 weeks more, leading to a shift in the mean seasonal cycle. The variability in this phase is much larger in the model, however, but the mean slope of melt matches the observations fairly well. At the Järnäs site the initial growth phase matches the observations better, with a similar growth rate and variability. The melt rate is also in line with the observation, however, there is still a shift in the mean seasonal cycle with the melt phase occurring a few weeks later in the season.

Here we used the cell-averaged thickness in the comparison with observations. Using the proxy level ice thickness (not shown) yields a slight underestimation at the Ratan sites and a larger underestimation at the more southern Järnäs site. For these coastal stations, which are just within the fast ice mask, it is evident that there is a signal of ridged ice late in the season affecting the seasonal cycles. Just as the ice breaks up and starts melting, small concentrations of relatively thick ice are usually advected into the fast ice zone, and thus strongly influence the variability. Our results also demonstrate that it is not straightforward to compare simulated grid point values with point measurements, mainly due to the difference in scales. The observations represent the ice thickness for one point in space, while the cell-averaged ice thickness in a model grid cell represents the mean of the thickness distribution on the scale of $\sim$4 km. Another contributing factor to the difference in the seasonal cycles is that the measurements rely on a stable ice cover. When the ice starts to form or break up it is much harder to go out and measure the thickness of the ice, thus the observations at the coastal stations Ratan and Järnäs underestimate the length of ice season.

## 3.5 Daily sea-ice extent and thickness for two extreme winters

To evaluate the model performance on shorter time scales we now briefly compare two single days from the hindcast with observational data from IceMap. We chose the day of the MBI for a mild winter (1995), and an extremely severe winter (1987), see e.g. Fig. 8. In Fig. 15 we show the extent and level ice thickness for NEMO-Nordic and IceMap, where we chose the date of MBI in the IceMap data set for both sources. As seen in Fig. 15 the extent of NEMO-Nordic's ice cover agrees well with the IceMap ice cover. For the mild winter the MBI from IceMap was $64\cdot10^3$ km$^2$ and occurred on the 16th of February. For NEMO-Nordic the total ice extent for the same date was $69\cdot10^3$ km$^2$, but the seasonal maximum was somewhat larger ($95\cdot10^3$ km$^2$) and reached already five days earlier. For the extremely severe winter the MBI from IceMap was $369\cdot10^3$ km$^2$ and reached on the 5th of March. Here the NEMO-Nordic total ice extent was $353\cdot10^3$ km$^2$, while the maximum simulated extent of $377\cdot10^3$ km$^2$ was reached nine days later. We note that there is an offset for NEMO-Nordic both in the total size of the sea-ice extent and the time when it occurs for these two cases. The IceMap data is updated roughly two times per week, which could partly explain the offset. When the model is run in forecast mode data assimilation of SST and sea-ice concentration will also likely improve the performance of the model, in terms of timing of the MBI furthermore.

For the mild winter the level ice thickness in NEMO-Nordic and IceMap agree quite well with the thickest ice found in the northern parts of Bothnian Bay, the eastern part of the Northern Quark and the eastern part of Gulf of Finland, while thinner ice is present in the central parts of the Bothnian Bay and along the coast of the Bothnian Sea. The area-averaged simulated ice thickness is somewhat thinner than the IceMap data. For the Bothnian Bay, Bothnian Sea and Gulf of Finland the NEMO-Nordic (IceMap) area-averaged thicknesses are: 26 (29), 10 (14) and 21 (23) cm. For the extremely severe winter, on the other hand, only the broad scale features are similar between the two. NEMO-Nordic generally has thicker ice compared to IceMap except for in the northern parts of Bothnian Bay. The area-averaged NEMO-Nordic (IceMap) ice thicknesses for the Bothnian Bay, Bothnian Sea and Gulf of Finland are: 59 (54), 50 (31) and 53 (50) cm. It is also evident in Fig. 15 that the IceMap thickness data is very patchy and only represent the large-scale features of the ice pack.

## 4    Summary and conclusions

We have presented the ice component of a new NEMO–LIM3.6 based configuration of the Baltic Sea. The model system is intended to be used for both climate studies and short-term forecasting in the Baltic Sea region. To adapt NEMO-Nordic to the Baltic Sea a number of parameterizations were tuned to the brackish Baltic Sea conditions. Compared to, for instance, polar regions this means that we had to tune the model to the Baltic sea ice which is only seasonal, thinner and has a much lower brine content. In addition, we implemented a simple fast ice parametrization which is based on the ocean depth.

In the present study we evaluated the performance of the model by comparing results from a 45-year long hindcast simulation, forced by data from a downscaled ERA-40 simulation, with several observational data sets. Most of our metrics are based on long-term changes in standard sea-ice parameters. However, we also briefly show how the model performs on a daily time scale by comparing daily means of the sea-ice state during the day of maximum extent for an extremely severe and mild winter.

Our results show that the NEMO–LIM3.6 modeling system is a tool well-suited to be used in a regional setup of a seasonally ice cover brackish Baltic Sea. The sea-ice concentration and extent are generally well simulated, when compared to the BASIS data set over the 1961–1979, although there is a bias in Kattegat, which might be related to a cold temperature bias in the air temperature forcing and the proximity to the open boundary. Some years there is also problems with too much ice in the Bothnian Bay. The downward trend of the MBI, over the 1961–2006 period, is much lower in NEMO-Nordic compared to the observational estimate. This is mainly related to the general overestimation of the ice cover in the two problem areas, which is also evident in the estimated SST biases. For the seasonal cycle of total ice extent we see an overestimation both during the growth phase and the melt phase, where the latter leads to a slightly faster ice retreat. The sea-ice thickness overall also agrees well with the observational data set. For the investigated period (1961–1979) the large-scale pattern shows good agreement between our level ice thickness estimate and the observations. The area-averaged thicknesses and area-integrated volumes of level ice are within the uncertainties of the observational estimates, also here we see an offset in the seasonal ice volumes similar to the ice cover offsets. From the mean total ice and snow thickness calculated from the ice thickness distribution, and the seasonal ice thickness at one out of two selected stations along the Swedish coast we see that the model has a tendency to underestimate the level ice thickness relative to the observational data sets. This is somewhat contradictory to what the negative

SST biases imply. For the mean total ice and snow thickness estimate based on the ice thickness distribution we speculate that a too thin snow cover can partly explain the difference. In our analysis we investigated biases in 2 m air temperature, snow thickness and SSTs, however, as we lack a complete set of observational estimates on all the different heat flux components at the ice–ocean–atmosphere interfaces we fail to attribute the driving causes of these anomalies.

For the demonstrated extremely severe and mild cases of day of MBI, the total extent and spatial distributions are well in line with the observational estimate, although there is an offset in total extent and when the day occurs in the model. Here the estimated level ice thickness is generally overestimated by the model for the extremely severe case while the mild case agrees better with the observational data. However, the observations which are weekly ice charts are very patchy and only represent the large-scale features of the ice. In addition, data assimilation of SST and sea-ice concentration will presumably improve the model skill further.

Furthermore, based on data from a few recent years our study shows that it is challenging to accurately capture the Baltic Sea ice thickness distribution. The simulated ice thickness distribution shows a more bimodal distribution compared to the observational data set. A higher number of categories as well as shifting the resolution towards thinner or thicker ice was explored, but failed to improve the distribution. Here we suggest further fine tuning of the ridging parameters and development of new transfer functions as a way forward to improve our model. However, we find that with the present ice category configurations the thickest ice category can be used as a proxy for ridge ice concentration and the lower four as proxy for level ice thickness and coverage. The proxy ridged ice concentration is generally in line with the observations, and the new fast ice parametrization yields a more realistic distribution with the ridges further off the coast.

The lack of reliable long-term and spatial representative observational data in ice covered regions limit our study to fully explore the model performance. Here particularly more snow and ice thickness data, as well as, radiation data would greatly improve future sea ice model assessments in the Baltic Sea. In addition, we caution that the observations, which are digitized hand drawn ice charts based on ship observations and various other sources, should be interpreted with some caution, as it is difficult to accurately estimate the ice thickness and deformation using that methodology. For future evaluations more objective methods should be preferred.

Finally, we have implemented a very simple fast ice parametrization, which is fixed based only on the ocean depth. For climate studies and forecasting purposes a more sophisticated parametrization is needed to capture long-term and seasonal changes in the fast ice zone. Modeling of the fast ice zone has received relatively little attention but recent studies (Lemieux et al., 2015; Olason, 2016) have suggested new ways to parametrize the fast ice zone, which could be feasible for a Baltic Sea ice model.

## 5   Code and data availability

NEMO-Nordic builds on the standard NEMO code (nemo_v3_6_STABLE, revision 5628) with only minor changes including: the fast ice parametrization and a spatial varying background viscosity/diffusivity that could be read in from file. The standard NEMO code can be downloaded from the NEMO web site (http://www.nemo-ocean.eu/). The nemo_v3_6_STABLE version

is available from the following link: http://forge.ipsl.jussieu.fr/nemo/svn/branches/2015/nemo_v3_6_STABLE. The new code blocks that are introduced (relative to the standard NEMO code nemo_v3_6_STABLE, revision 5628) into our NEMO-Nordic code are included as supplemental material. The full NEMO-Nordic code is in a Subversion revision control system repository, available under http://54.73.141.37/subversion/repository/source_code/trunk/NEMOGCM. However, a user account is needed

5   to gain full access. This work used the revision 339 of NEMO-Nordic code. Access to the NEMO-Nordic code and all input data, analysis scripts and data used to produce the figures in this study can be made available upon request by the corresponding author.

*Acknowledgements.* This research is part of the project BONUS STORMWINDS with financial support from BONUS, the joint Baltic Sea research and development programme (Art 185), funded jointly from the European Unions Seventh Programme for research, technological

10   development and demonstration and from the Swedish research council for environment, agriculture sciences and spatial planning (FOR-MAS). This work is also partly funded by the SmartSea project of the Strategic Research Council of Academy of Finland, grant No: 292 985. We thank the Ice Service at SMHI as well as Christian Haas for providing data and the NEMO-LIM3 developers for their efforts in developing the code.

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

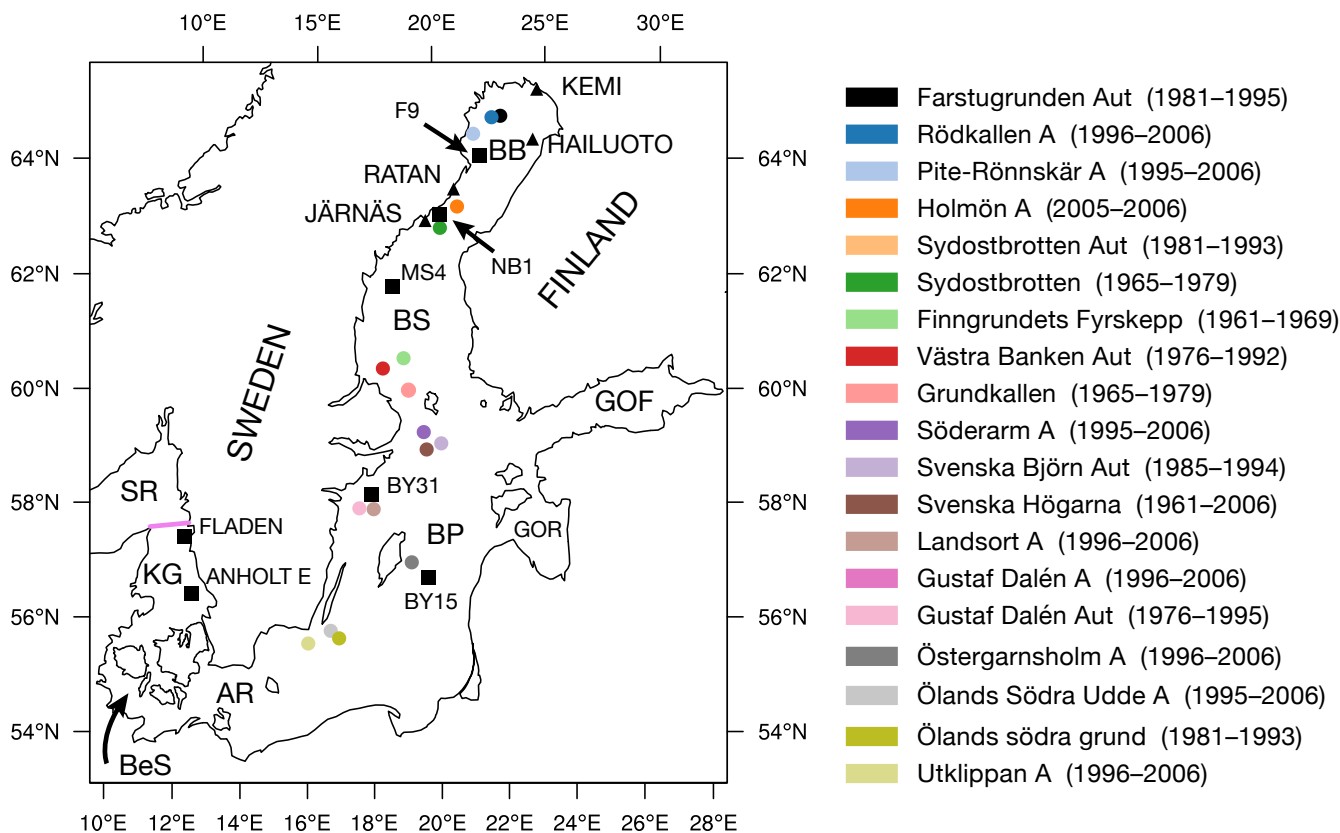

**Figure 1.** Map showing names of places mentioned in the text; the sub-basins are abbreviated: BB=Bothnian Bay, BS=Bothnian Sea, GOF=Gulf of Finland, GOR=Gulf of Riga, BP=Baltic proper, GR=Gulf of Riga, KG=Kattegat, AR=Arkona Sea, BeS=Belt Sea and SR=Skagerrak. The pink line shows the open boundary in Kattegat; the black triangles show the coastal sites Hailuoto, Kemi, Ratan and Järnäs; and the black squares show the Fladen, Anholt E, BY15, BY31, MS4, NB1, F9 stations. The air temperature stations are depicted by the dots color coded according to the legend to the right; the time period for each air temperature station is shown within the brackets.

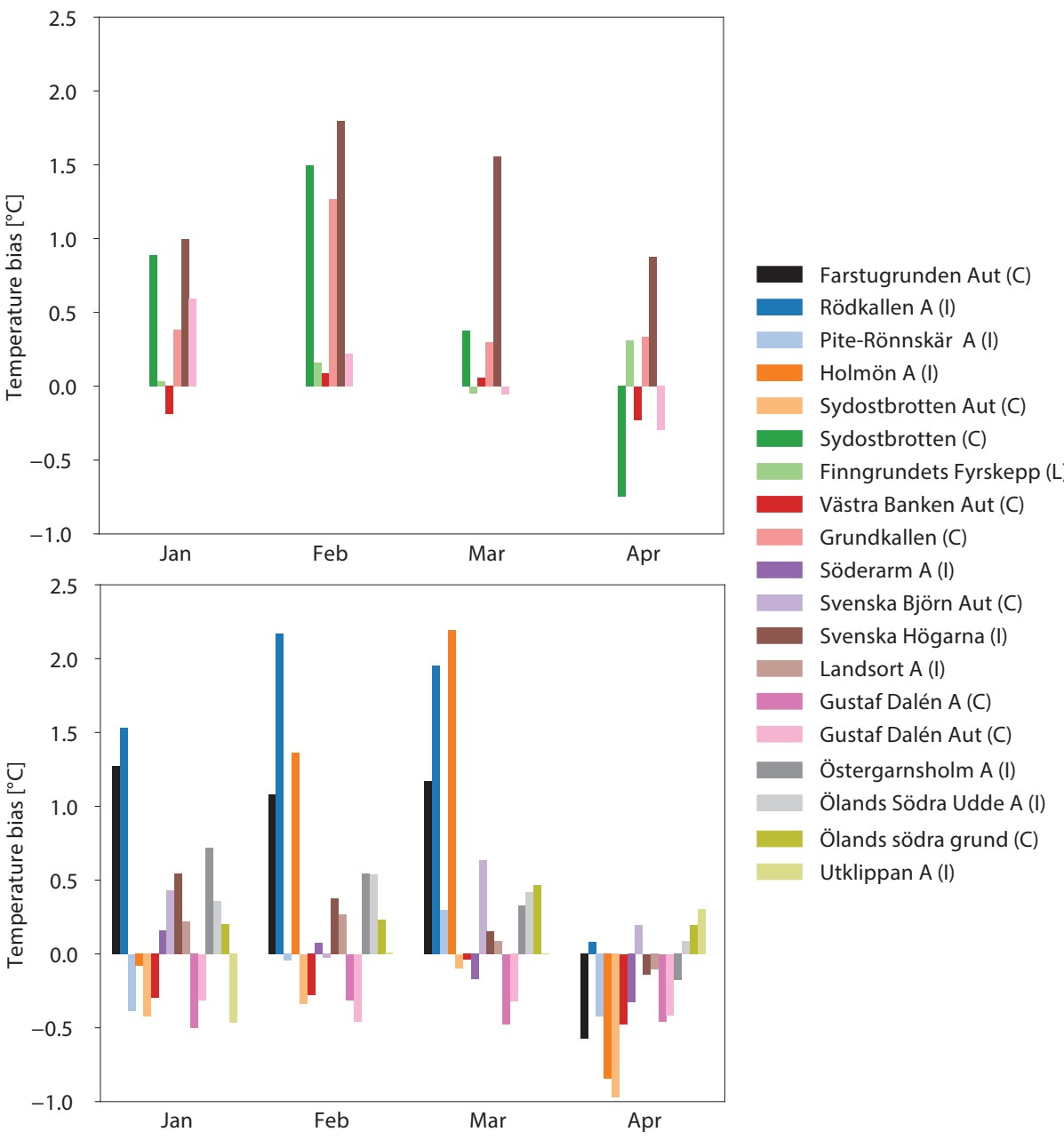

**Figure 2.** Long-term monthly mean January–April 2 m air temperature bias ($^\circ C$). The bias is calculated for before 1979 (left) and after 1979 (right), the stations are arranged in an approximately north to south order going from the leftmost to the rightmost bar in each month. A positive bias means that the model air temperature forcing is warmer than the observations. After each name the type of measurement platform is given: C=Cassion Lighthouse, L=Lightship, I=Island.

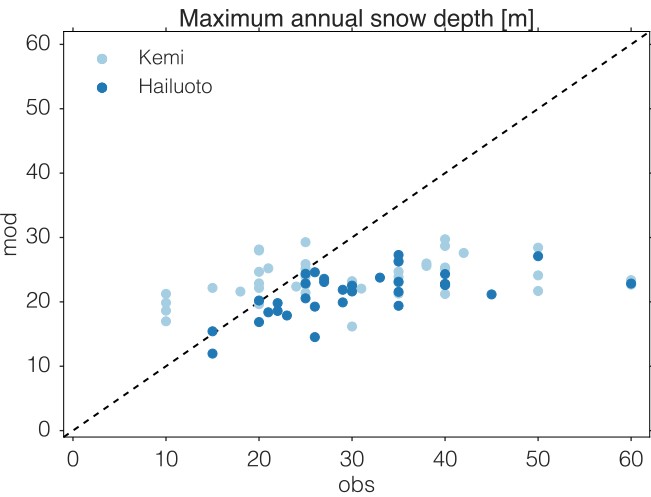

**Figure 3.** Comparison between observed and simulated maximum annual snow depth at two stations (Kemi and Hailuto, see Fig. 1) in the Bothnian Bay. Observed snow depths are on the x-axis and NEMO-Nordic snow depths on the y-axis. The Hailuoto station covers the period 1974–2006 and the Kemi station the period 1961–2005.

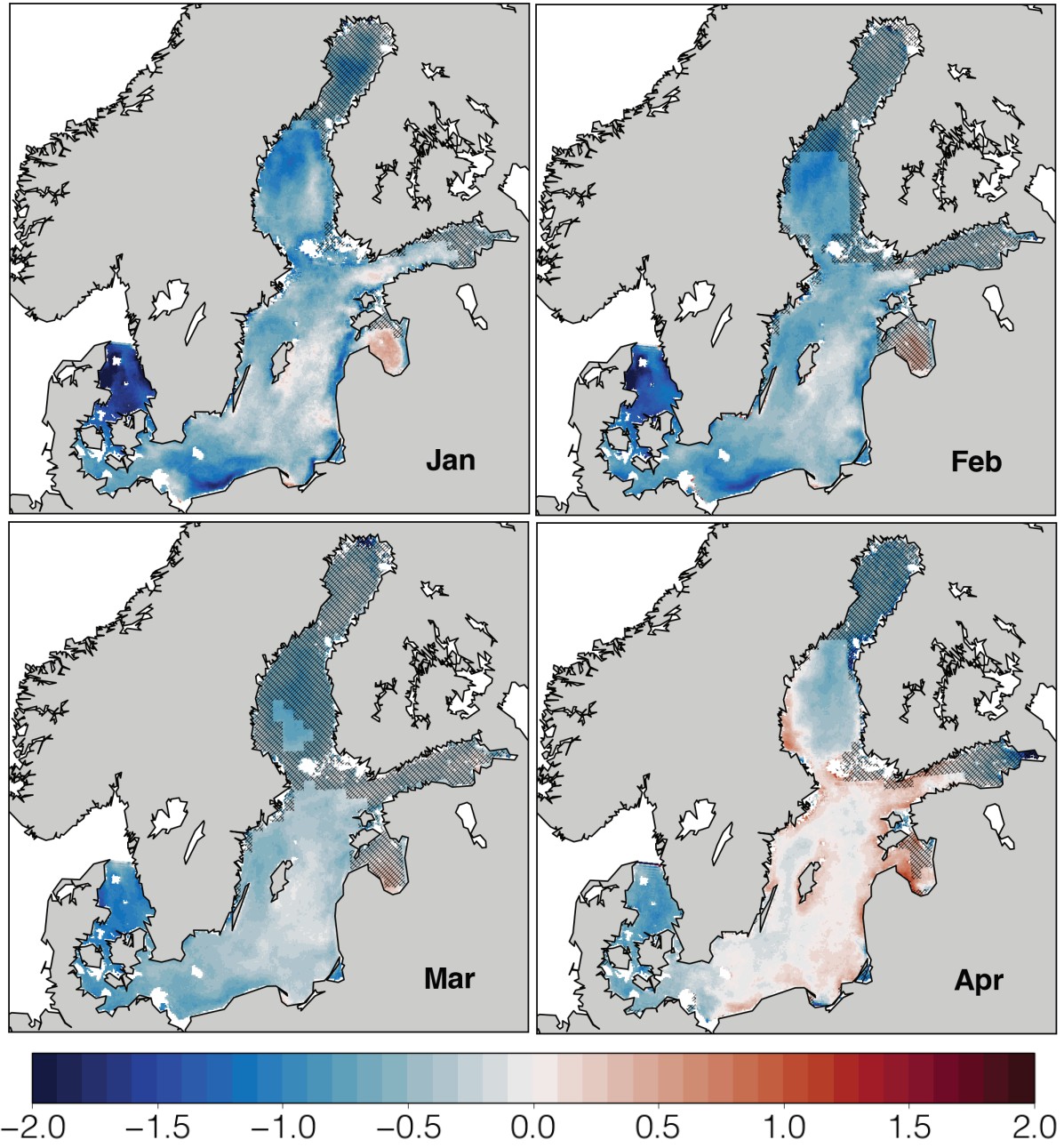

**Figure 4.** Long-term monthly mean January–April SST bias ($^{\circ}C$) for the period 1990/91–2005/06. Positive values mean that NEMO-Nordic is warmer than the BSH satellite observations. Areas where the BASIS/IceMap ice concentration is greater than 15% have been hatched.

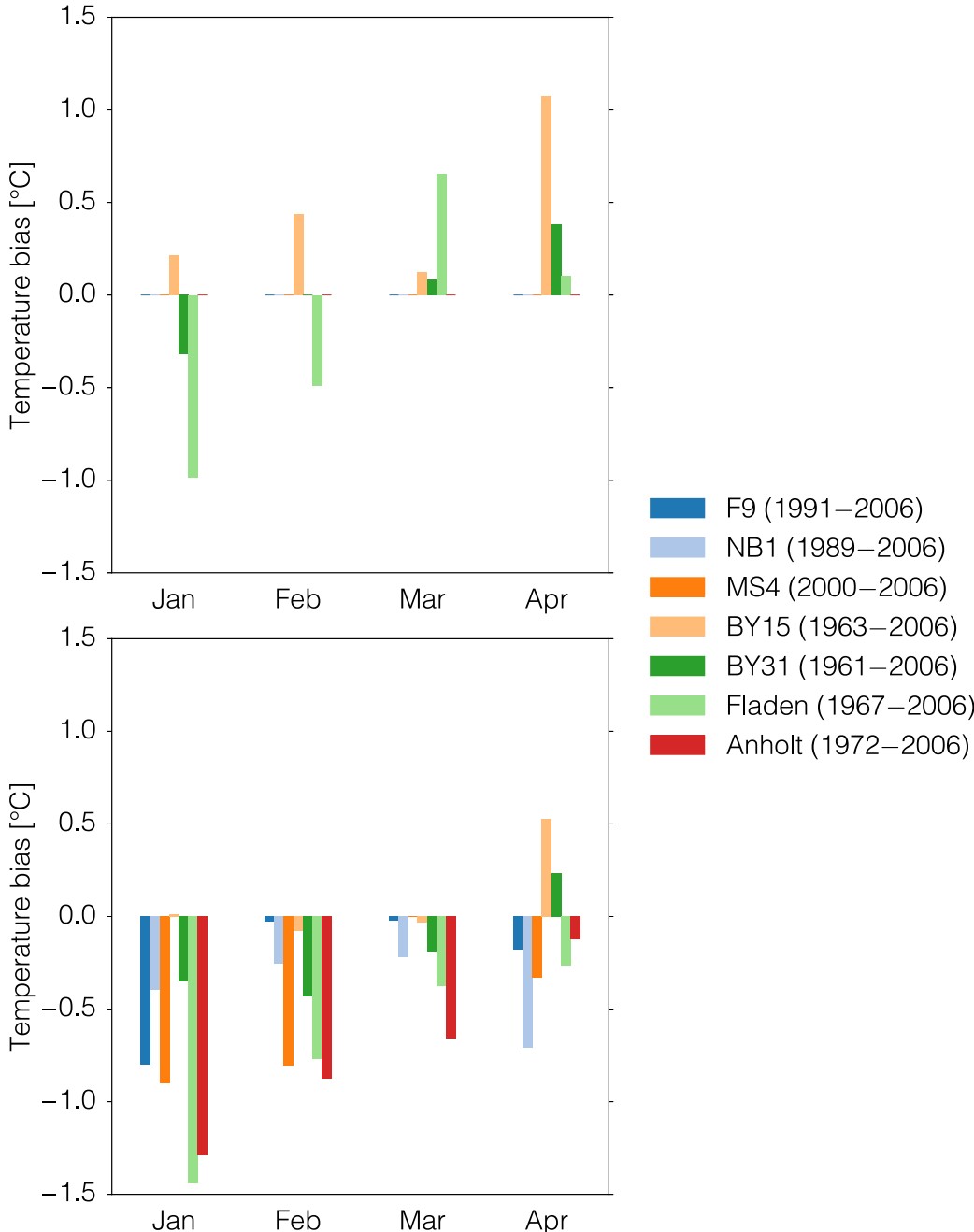

**Figure 5.** January–April SST biases for a selected number of stations (arrange in north–south order) estimated from CTD casts. The upper panel shows biases for stations available before 1979 and the lower panel after 1979. The full time period of each station is given in the figure legend; for the location of the different stations see Fig. 1. Positive values mean that NEMO-Nordic is warmer than the CTD casts.

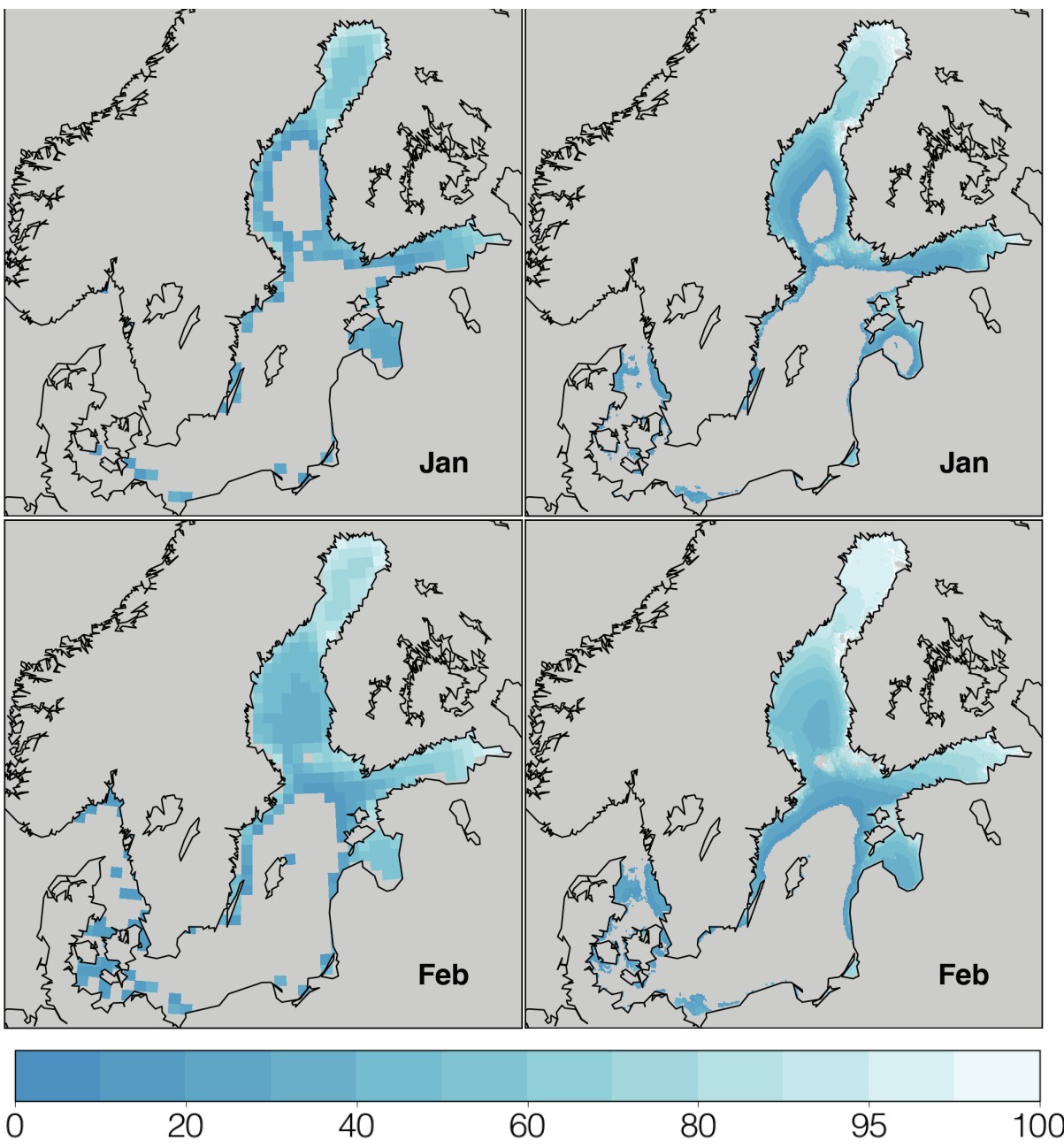

**Figure 6.** Climatological mean ice concentration for January and February, for the period 1961/62–1978/79. The left column shows sea-ice concentration for BASIS and the right for NEMO-Nordic. Note that grid cells with an ice concentration lower than 15% have been masked out and that grey sea areas denote missing or masked values.

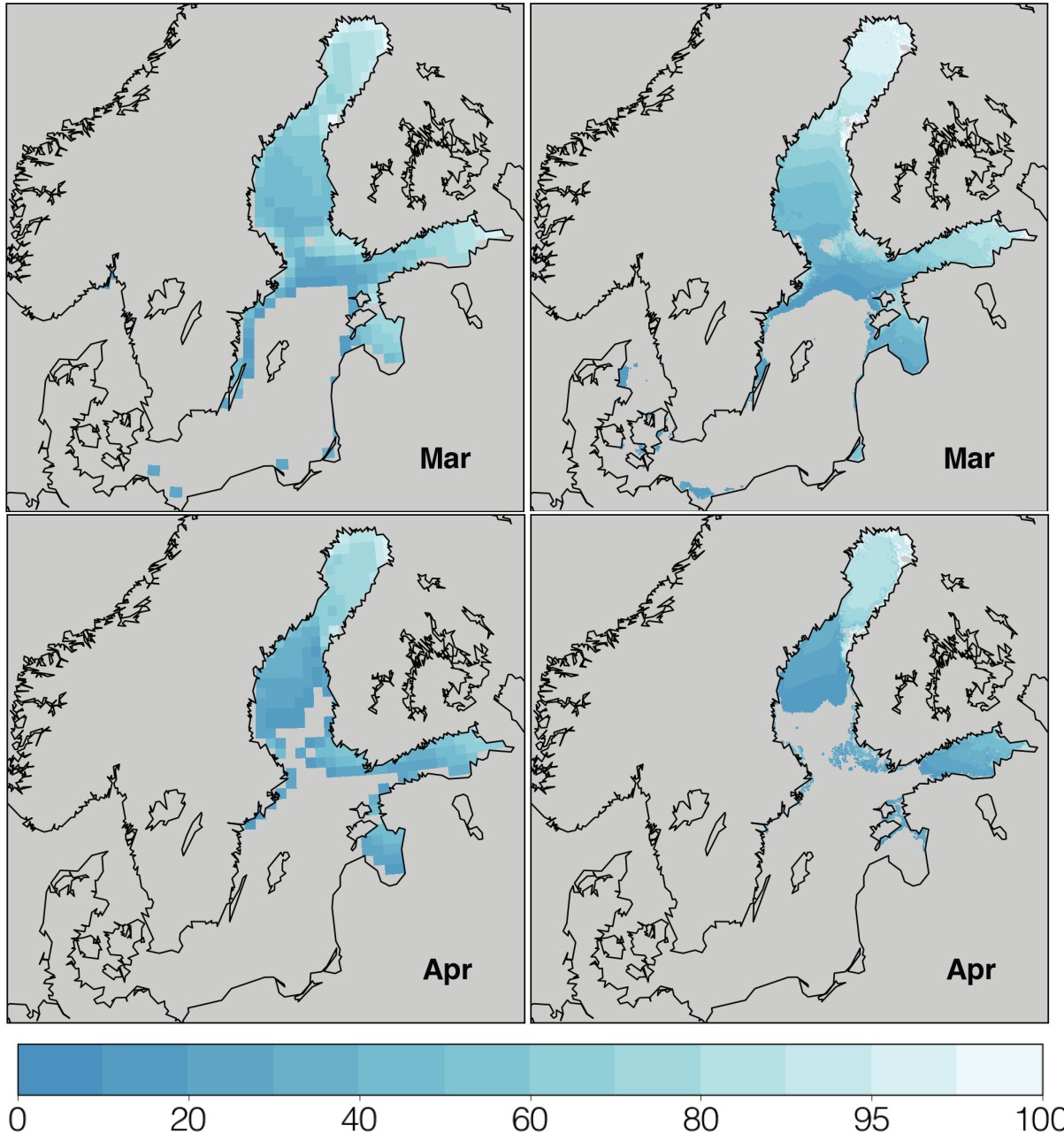

**Figure 7.** Same as Figure 6 but for March and April.

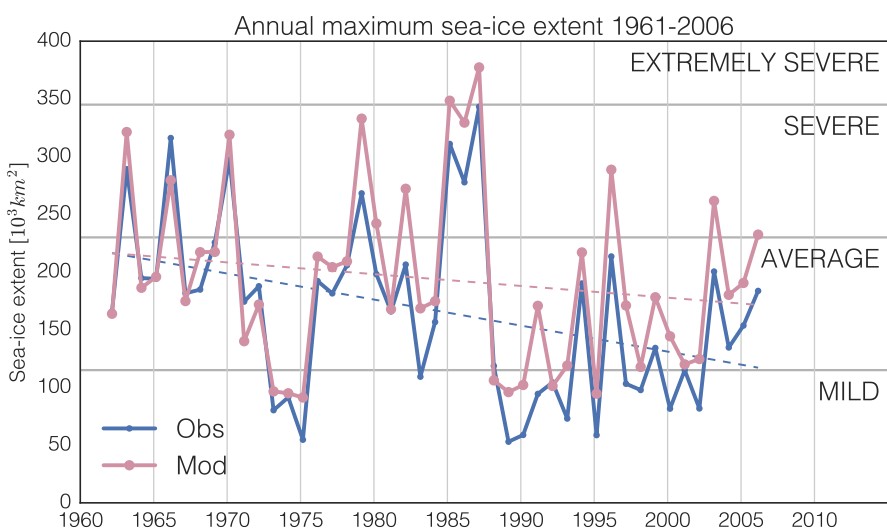

**Figure 8.** Simulated (red solid line) and observed (blue solid line) annual maximum daily sea-ice extent in the Baltic Sea, for the period 1961/62–2005/06. The horizontal gray lines show the limits for winters with a maximum sea-ice extent in the Baltic Sea classified as mild, average, severe and extremely severe, respectively. The dashed red and blue lines show the linear trends for simulated and observed maximum sea-ice extent, respectively.

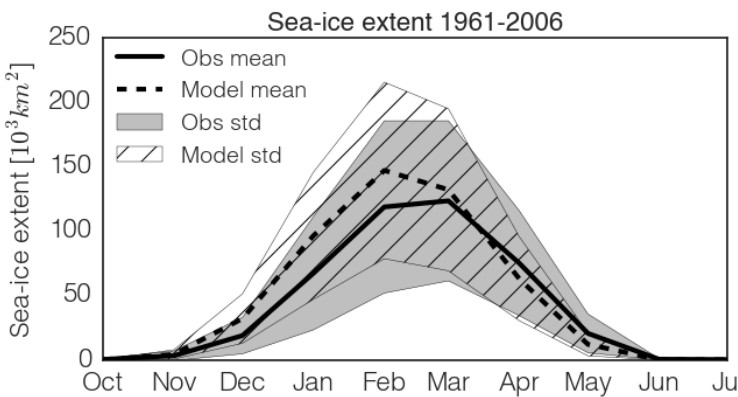

**Figure 9.** Seasonal cycles of simulated and observed total sea-ice extent in the Baltic Sea. The lines show the seasonal monthly means and the filled envelopes show the ± 1 standard deviations. The seasonal cycles are calculated for the periods 1961/62–2005/06. Note that Skagerrak has been masked out in the observational data set.

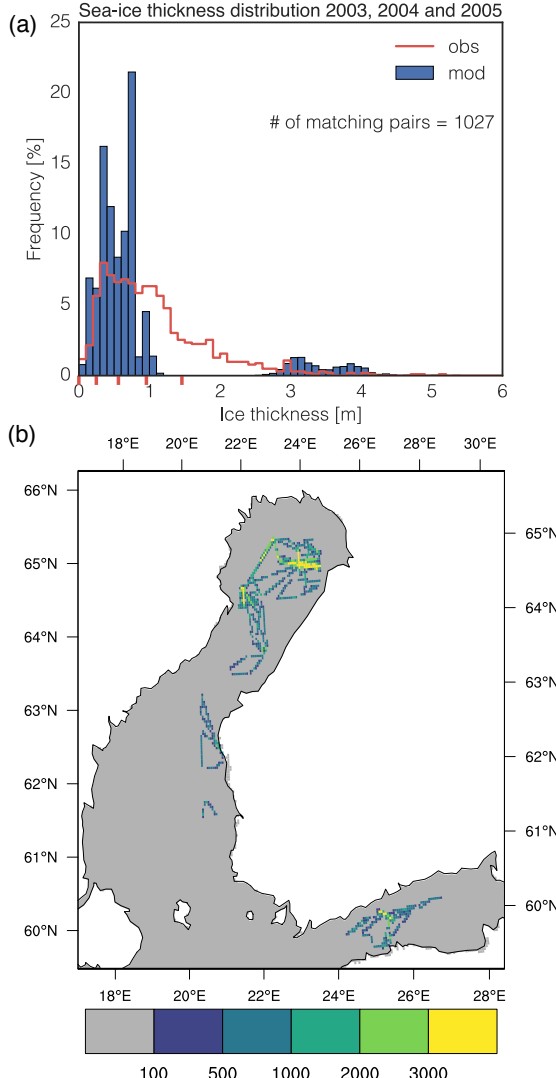

**Figure 10.** (a) Sea-ice thickness distributions calculated from the gridded EM data (blue) and the simulation (green). (b) shows the number of sub-grid observations of the gridded EM data. The red tick lines in (a) show the limits for the five ice categories, note that the upper limit of the last ice category – which is unbounded – is not shown.

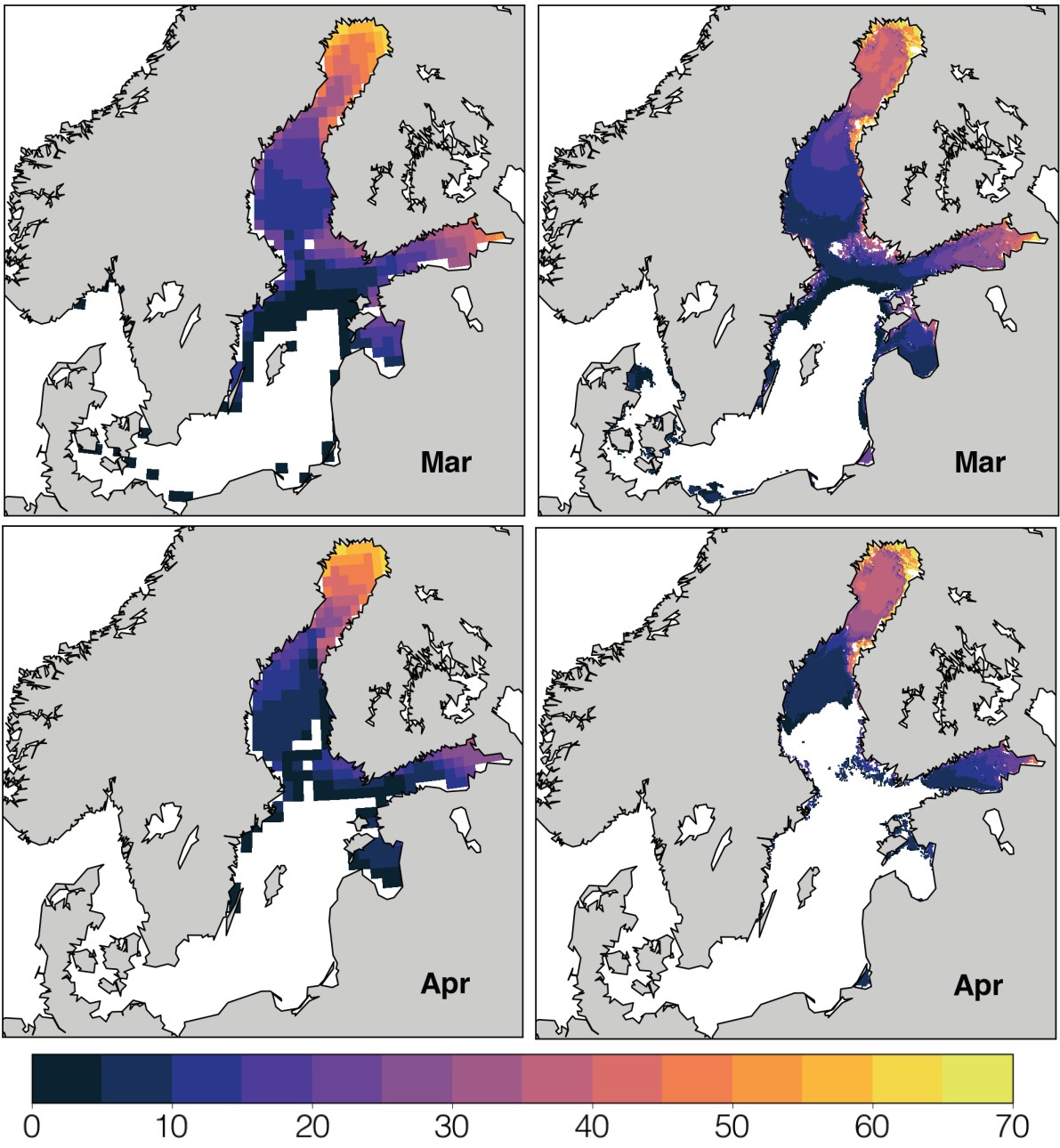

**Figure 11.** Climatological mean sea-ice thickness in cm for March and April, for the period 1961/62–1978/79. The left (right) column shows sea-ice thickness for BASIS (NEMO-Nordic). For NEMO-Nordic the mean sea-ice thickness represents a proxy level ice thickness and is calculated using a category-weighted average of the first four ice categories, see Eq. 4. Note that grid cells with an ice concentration lower than 15% have been masked out.

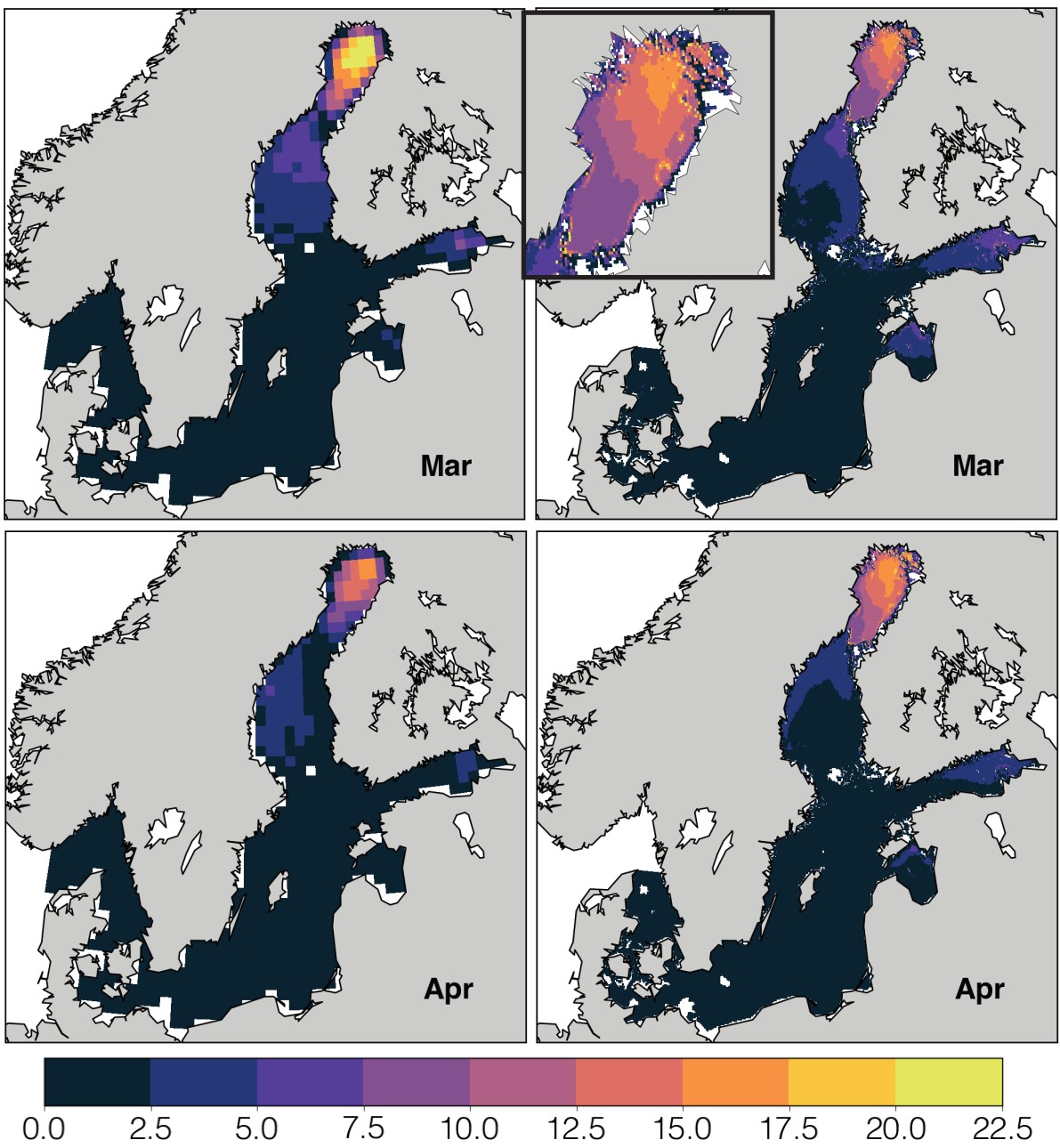

**Figure 12.** Climatological mean sea-ice ridge concentration in % for March and April, for the period 1961/62–1978/79. The left (right) column shows sea-ice ridge concentration for BASIS (NEMO-Nordic). For NEMO-Nordic the sea-ice ridge concentration is estimated from the fifth ice category. The inset shows a zoom over the Bothnian Bay for the simulated March distribution.

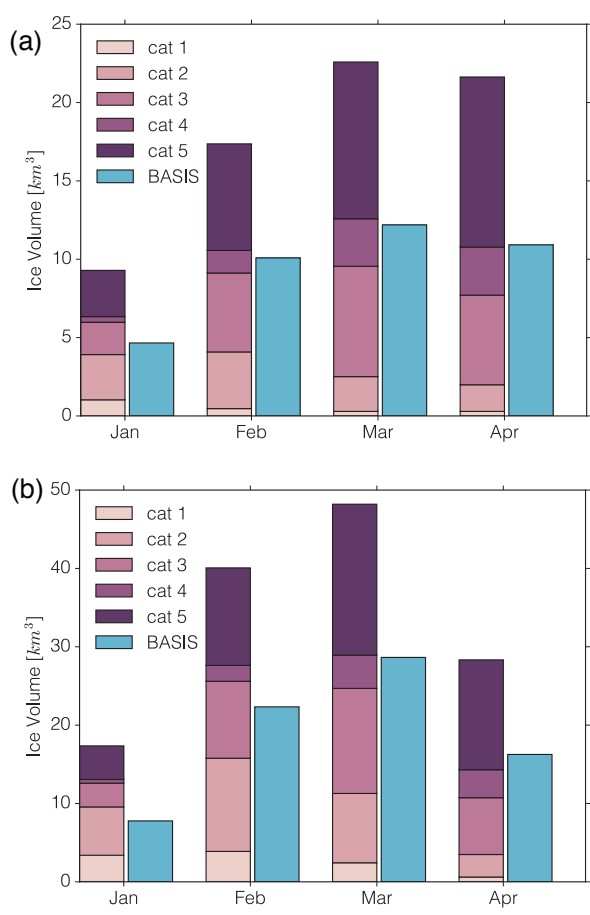

**Figure 13.** Climatological mean sea-ice volume for January–April, for the period 1961/62–1978/79, calculated for (a) the Bothnian Bay and (b) the entire Baltic Sea. Note that the NEMO-Nordic ice volumes are calculated for each ice category and the sum gives the total simulated ice volume.

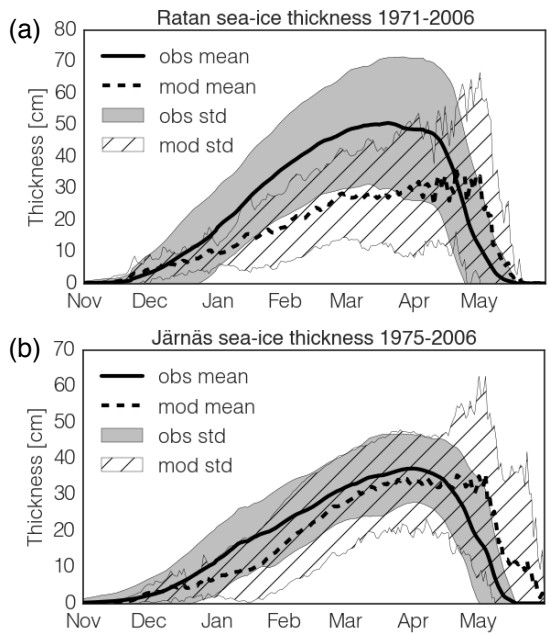

**Figure 14.** Seasonal simulated and observed sea-ice thickness at the coastal sites (a) Ratan and (b) Järnäs. The lines show the seasonal daily means and the filled envelopes show the ± 1 standard deviations. The seasonal cycles are calculated for the periods 1971/72–2005/06 and 1975/76–2005/06 for the Ratan and Järnäs sites, respectively. For the location of the sites see Fig. 1.

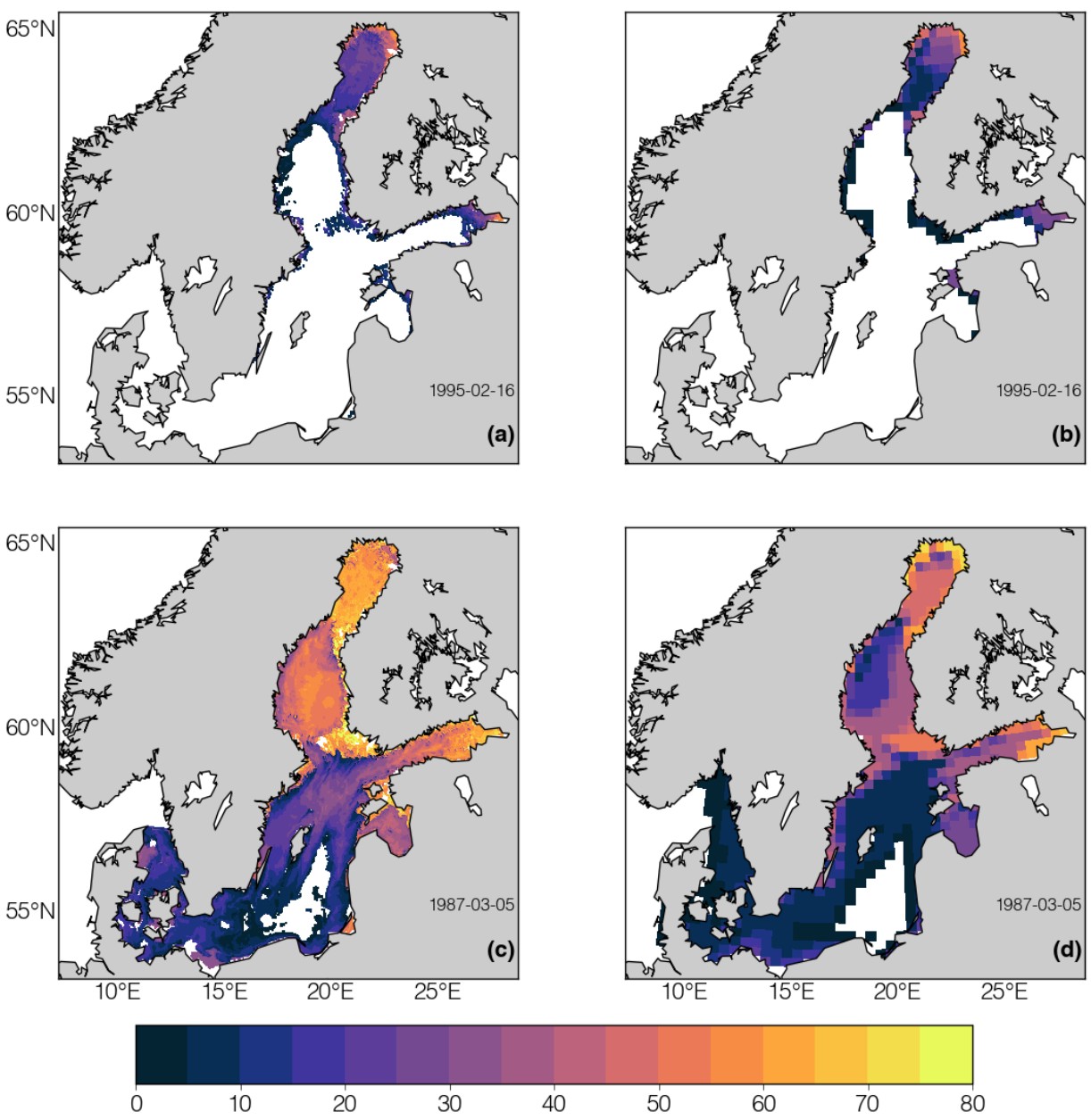

**Figure 15.** Daily means of level ice thickness based on IceMap data (right column) and proxy level ice thickness from NEMO-Nordic (left column) for the day of MBI for: a mild winter 1995-02-16 (upper row); and an extremely severe winter 1987-03-05 (lower row). Note that grid cells with an ice concentration lower than 15% have been masked out.

**Table 1.** Physical parameters in the sea ice namelist (namelist_ice_ref) that where changed in NEMO-Nordic compared to the standard global ORCA2-LIM3 that is include in the NEMO–LIM3.6 model system.

| Namelist parameter | NEMO-Nordic | ORCA2-LIM3 | Unit | Description |
|---|---|---|---|---|
| rn_hicemean | 0.5 | 2.0 | $m$ | Expected domain-average ice thickness, value based on observational studies (Vihma and Haapala, 2009). |
| rn_pstar | 2.5e+4 | 2.0e+4 | $Nm^{-2}$ | Ice strength thickness parameter, value based on previous modeling studies (Leppäranta et al., 1998). |
| rn_ ahi0_ref | 1.0 | 350.0 | $m^2s^{-1}$ | Horizontal sea ice diffusivity, only for numerical reasons, selected as low as possible. |
| rn_hnewice | 0.01 | 0.1 | $m$ | Thickness for new ice formation in open water, a low value to capture thin new ice formation. |
| rn_maxfrazb | 0.0 | 1.0 | | Maximum fraction of frazil ice collecting at the ice base, neglected in this Baltic Sea setup. |
| rn_himin | 0.01 | 0.1 | $m$ | Minimum ice thickness used in remapping, a low value to capture thin new ice formation. |
| rn_betas | 1.0 | 0.66 | $m$ | Exponent in lead-ice repartition of snow precipitation. |
| rn_icesal | 1.0e-3 | 4.0 | $g\,kg^{-1}$ | Bulk sea-ice salinity, chosen for numerical reasons, set to minimum value. |
| rn_hstar | 30.0 | 100.0 | $m$ | Determines the maximum thickness of ridged ice, value based on observations (Vihma and Haapala, 2009). |
| rn_hraft | 0.07 | 0.75 | $m$ | Threshold thickness for rafting, value based on analytical modeling (Parmerter, 1975). |