# Peer review of "Sea-ice evaluation of NEMO-Nordic 1.0: a NEMO-LIM3.6 based ocean—sea ice model setup for the North Sea and Baltic Sea"

_Geoscientific Model Development, 2017_

## Short Comment (SC1) · 3 Apr 2017

Review of Pemberton et al.: "Sea-ice evaluation of NEMO-Nordic 1.0: a NEMO–LIM3.6 based ocean–sea ice model setup for the North Sea and Baltic Sea"

by F. Massonnet

The study presents the sea-ice component of NEMO-Nordic, a new regional configuration based on of the European community ocean-sea ice model NEMO. The authors have adapted the Louvain-la-Neuve Ice Model LIM3 to match the specificities of the Baltic Sea, and run a control hindcast of 45 years (1961-2006). The model output is compared to data from ice charts. The model is found to be in overall good agreement with observational data, but to underestimate the volume of level ice and the sea surface temperatures.

To my point of view the study is adapted for Geoscientific Model Development, as it presents the final product obtained after certainly a lot of developments (I acknowledge the work of creating a new configuration). On the scientific side, I find the paper very descriptive, and sometimes speculative, and would therefore welcome more discussion or experiments to understand the origins of specific results (I'm giving a few suggestions below)

I appreciate the effort to develop this configuration as part of a community effort (NEMO) and believe that the results are satisfactory enough to publish this study. However, I have two main points that I would like the authors to make clearer, possibly by running one or two additional short experiments:

1. By prescribing a constant sea ice salinity of 0.001 PSU in the model, the authors essentially switch off the halo-dynamics in the sea ice model. What is the reason that led them to that choice? Did the authors conduct tests with the full interactive halo-dynamics module of LIM active, and compared the results? I know that the Baltic sea is a particularly fresh sea, hence the ice is mostly non-salty, but why not try to let the model figure it out itself?

2. The authors diagnose the proportion of ridged ice as the ice concentration in the thickest category (p. 5, lines 16-21). The authors correctly write that this is an approximation. I'm wondering if the authors could go a bit more quantitative here. For example, by running the same simulation but deactivating the ridging/rafting scheme, they could measure the amount of thermodynamically-grown ice in the fifth category. If that amount is very small (say, < 5 %) then indeed that category in the reference run contains most of the deformed ice.

Other comments: - How did the authors come to the specific values of LIM for table 1? How did they tune this configuration? Providing a bit of background, with lessons

learned, would be very helpful to those authors that may want to run their own configuration (e.g. in Hudson Bay).

- Given the high resolution of ~4 km, the continuum hypothesis for the sea ice medium is not necessarily true. The (elastic) viscous-plastic rheology used in LIM3 heavily rests on that assumption, though. Did the authors experience numerical instabilities? Did they check the sea ice velocity/deformation fields? What would be their advice in terms of rheology at that high resolution?

- What are the ocean and sea ice model time steps?

- Fig. 12 suggests that the simulated volume overestimates the observational reference BASIS. However, as explained in the text, BASIS is essentially a measure of level ice volume. Given the fact that the atmospheric forcing is already cold, the authors conclude that the simulated volume is definitely too low. I have two questions here: 1) Are there uncertainties available around the BASIS estimates? 2) What does the mismatch in simulated and observed volume tell about the sea ice model biases? If I understand, this is more a thermodynamic issue since the deformed ice was not considered. How would the authors do to investigate the origin of this bias? Is it possibly related to the conductivity of snow for which could be significantly different from the one used for the Arctic? See also my next comment

- In the conclusion, the authors note that the cold bias in SST in the model is somehow contradictory with the negative bias in sea ice volume, and point towards the possible role of snow. I assume that the snow conductivity was set to its default value. However, is there a good reason to assume that snow conductivity in the Baltic Sea is the same as that of the polar regions?

Minor comments

p. 1, l. 16: "... the ice extent can reach a coverage of almost 100% during severe winters". Ice extent has usually units, which makes this sentence confusing. Consider

"ice coverage reaches 100% during severe winters"

p. 3, l. 19: To how many vertical levels does the configuration correspond? Please specify.

p. 4, l. 5: "include" –> "included"

p. 4, l. 8: "uses" –> use

p. 4., l. 26. I don't understand the sentence saying that the scale of most sea-ice models is ∼ 1 km. Most large-scale sea ice models currently run at ∼10 to ∼50 km, which is at least one order of magnitude larger than what is written.

p. 5, l. 27: "show" –> shows

p. 7, l. 25: An extra "the" is present.

Fig. 4. The color bar is misleading because "100% ice" and "no ice" have essentially the same colour, i.e. white. Consider a more adapter colormap.

Fig. 8a. The figure 8a seems to suggest that the sea ice model grows too much ice in the thin categories. Did the authors try to run sensitivity tests with difference ice thickness boundaries? It would be interesting to at least identify possible candidates to explain the striking differences seen in Fig. 8a. This is somehow done in the text (p. 10, first paragraph) but in a very descriptive and speculative way.

---

## Referee Comment (RC2) · D. Bailey (Referee) · 6 Apr 2017

Overall this was an interesting description of an ice-ocean modelling system for the Baltic Sea. I do have a few concerns, but feel that the manuscript may be acceptable after some relatively minor revision. There were no scientific hypotheses in this paper, so I am simply evaluating the technical detail aspects.

1. My first concern is about the model boundary conditions. I understand that this domain is forced from a larger North Atlantic domain and that the ocean exchange is not a key part of the Baltic Sea ocean temperature and salinity structure. However, what happens with fresh water from the atmosphere and land? What impact does the river discharge have on the sea surface temperature and salinity of the Baltic? How is

this specified in the modelling system?

2. Despite the SST biases as described in the manuscript, the model does a pretty nice job at simulating the sea ice concentration, thickness, and extent. I wonder about the snow on the sea ice in this region? I know the authors present a climatological seasonal cycle of snow from the model, but I would like to see more discussion here. Can the authors add the observed snow depth information to Figure 15? Are there no other sites or model information to get snow depth? Maybe accumulating snowfall from the forcing? As the authors know, the albedo of the snow is critical to the seasonal evolution of the sea ice. Albedo formulation? Parameter settings?

3. One thing I don't know about the Baltic is whether the daily maximum sea ice extent always occurs in the same month? It is sometime in February or March. It is interesting in Figure 7 that the model maximum extent is in March, while the observations indicate February. There should be more discussion of this.

4. From Figure 8, it looks like the model has too much thin ice (less than 1m) and not enough thick ice. How does this vary from year to year? I wonder if there is a relationship to snow here. Is there a freeboard parameterization in the model? Are you getting to much or too little snow-ice formation?

5. Finally, there are a lot of figures (sixteen) given the amount of text in the manuscript. A number of the figures are very qualitative, just comparing contour plots by eye. Could some of these be condensed into more quantitative information, maybe spatial correlations or differences between the model and observations? Just a thought.

---

## Referee Comment (RC3) · M. Vancoppenolle (Referee) · 10 Apr 2017

The paper is based on good and original material, and deserved to be published. Writing is generally good. I found the paper material quite interesting.

Yet I believe it should be improved, following two categories of objections.

1) The conclusions could be more general and interesting.

2) The analysis of results could be more acute and precise

Besides, some sections (3.1, 3.4) are not fully clear and could be sharpened.

I think the manuscript can easily be improved, and I hope my comments will help.

[Figure]

\*\*\* General comment #1. The conclusions could be more generic.

I would somehow use the results of the paper paper to question the capabilities of regional modelling ice systems forced by atmospheric reanalysis, considering Baltic Sea ice as a successful example.

The following core of conclusions could be the base of the abstract/

Conclusion 1: The NEMO-Nordic ice modelling system is appropriate to get the mean extent, volume, and geographical distributions of ice concentration and thickness in the Baltic Sea, which all seem rather precisely captured (within 10% of obs?). The ice melts early, which is attributed to XXX.

Conclusion 2: Extreme years, in particular severe winters, are more difficult to simulate.

Conclusion 3: The subgrid scale ice thickness distribution seems challenging.

\*\*\* General comment #2. The analysis of results could be more acute and precise

- The SST bias should be analyzed with respect to ice presence or absence. In presence of ice, the SST must be very very close to the freezing point, hence any SST bias has to be attributed to SSS. The warm bias in April is very likely due the early ice retreat, which was barely mentioned in the text.

- Why ice melts too early is not clearly attributed. There are admittedly bits and pieces, but the analysis could be more systematic (air temperature, snow depth, incident solar radiation, surface albedo). The ice thickness bias is neither clearly quantified.

- The ice thickness distribution analysis would be more conclusive if (i) the model ice categories were used for both observations and model, (ii) the exact same time and locations were used to construct the pdf. At this stage, the model looks really bad, but this could be because different ice categories are used.

- The comparison of snow at two stations is not enough to conclude, as blowing snow effects can be locally dominant. As the present analysis is not meaningful, you could

remove it and just mention it as being inconclusive. Snow depth could be critical and I'm surprised there are not more snow depth observations to compare with, in such a well studied coastal sea. If available, a more systematic snow depth analysis would be a real plus.

- I think the FDD-model analysis is confusing and adds unnecessary complexity to the paper. I would recommend to compare the winter air temperature bias instead, that would be simpler and actually equivalent.

- The effect of fast ice parameterization is claimed to improve the results, but this is not supported by material. Therefore, I would recommend to be more explicit.

- The analysis of extreme years is quite interesting. It should be stated or shown whether the analysis applies to all extreme ice years. Do all severe years follow a similar ice thickness pattern? Are all mild years realistically captured. In addition, whether forcing or model are responsible should be at least discussed.

- Be more quantitative in the text in general (give numbers instead of "reasonable" or "quite good")

- Revise your conclusions once analysis has been sharpened.

*** Detailed comments

- Introduction can be sharper and a better selection of the required elements could be done.

- p4, l.5 "include"D"

- p4. l.17 give reference, because 0.17m is not the common value in LIM3. In practice, you removed rafting from the model. Why did you do that ?

- p.4 your fast ice parameterization is grid-size dependent because your criterion is based on cubic meter. Why did not you use a volume of ice per grid cell area criterion ?

- p.5 your mean thickness is now referred to as volume per unit area (Notz et al, TC 2016).

- p.5 line 14, why do you use 5 and not 4 in the denominator to compute your level ice thickness ?

- p.6 line 20. Could you describe in one sentence what is the observation based for ice charts ? satellite ? visual ? how was ice thickness quantified ?

- section 3.1 should probably be revised, I found it hard to follow (see general comment as well).

- p. 9, l. 9 "variability" -> "interannual variability". "quite well" -> be more quantitative. Are you sure the units of STD are correct ?

- p. 9 l. 13. What about the trend if you exclude the first 15 years ? May be use ice area to see if that is robust ?

- p.10 I can't reconcile the file that volume could be overestimated with the fact that the ice melts too early.

- p. 11, l 16 "consistS"

- section 3.4. If you keep this analysis, which I don't especially recommend, you may want to explain why you use different sites for FDD and for snow depth. I finally figured why, but it would have been good if you had said it right away.

- acknowledgements: acknowledge NEMO developers :-)

————————————————

---

## Editor Comment (EC1) · A. Yool (Editor) · 18 Apr 2017

Dear Authors,

The discussions phase of your manuscript "Sea-ice evaluation of NEMO-Nordic 1.0: a NEMO–LIM3.6 based ocean–sea ice model setup for the North Sea and Baltic Sea" is now complete.

There are three referee reports on your manuscript. All three find the manuscript publishable after revision, with almost all comments requesting clarification or expansion of the manuscript to address specific ambiguities or omissions. Referee 1 has suggested two key points (sea-ice salinity and ridged ice) that they believe may require further

simulations to be resolved to their satisfaction.

Please respond fully to all of the points raised by the referees. To facilitate this, I would be grateful if you could provide (a) a revised manuscript, (b) full responses to the referee comments, (c) some form of tracked-changes document to clearly identify manuscript revisions.

If you have any questions, please do not hesitate to get in contact.

With best regards,

Andrew Yool
* * *

---

## Author Comment (AC1) · 8 Jun 2017

We thank the topical editor Andrey Yool and the three referees Francois Massonet, David Bailey and Martin Vancoppenolle for their efforts reviewing our discussion paper. This is our authors final response to all the comments made by the referees. For our own convenience we gathered all replies in the same document (see supplemental pdf). Many thoughtful points where raised and the suggestions and comments, we believe, helped us improve the manuscript.

Following the suggestions, the following items have been explored/changed:

[Figure]

* We ran a suite of sensitivity experiments to: i) evaluate the thermodynamically vs dynamically grown ice in the fifth category. ii) evaluated the impact of the number ice categories and the distribution of category bounds on the ice thickness distribution. iii) estimated the amount snow-ice formation in the model.

* Evaluated the air temperature and snow thickness bias against a set of observations from island, lightships, caisson lighthouses and on-ice measurements.

* Removed the FDD section.

* Extended section 3.1.

* Regridded the BASIS/IceMap data to the same grid as the ocean model for a better comparison of the area-integrated and area-averaged quantities.

* Removed the FDD and seasonal snow thickness figures (old figs 14 and 15).

* Added new figures for air temperature (new fig 2) and snow thickness bias (new fig 3).

* Changed the SST bias figure (fig 5) to include more stations

A detailed account of how we have addressed the general and specific comments and how we propose to change the revised manuscript is provided as a supplemental pdf-document. Our comments are given in bold text, in the supplemental document. In addition, the pdf-document also contains a latex-diff with highlighted text changes inserted. We hope that our proposed changes and our answers to all the referees' questions and comments are to your satisfaction.

Kind Regards, Per Pemberton (on behalf of the co-authors)

Please also note the supplement to this comment:
http://www.geosci-model-dev-discuss.net/gmd-2017-10/gmd-2017-10-AC1-supplement.pdf

[Figure]

**Supplement:**

**Authors response to referees**

**Response to reviewer 1**

Review of Pemberton et al.: "Sea-ice evaluation of NEMO-Nordic 1.0: a NEMO–LIM3.6 based ocean–sea ice model setup for the North Sea and Baltic Sea"

by F. Massonnet

The study presents the sea-ice component of NEMO-Nordic, a new regional configu- ration based on of the European community ocean-sea ice model NEMO. The authors have adapted the Louvain-la-Neuve Ice Model LIM3 to match the specificities of the Baltic Sea, and run a control hindcast of 45 years (1961-2006). The model output is compared to data from ice charts. The model is found to be in overall good agreement with observational data, but to underestimate the volume of level ice and the sea surface temperatures.

To my point of view the study is adapted for Geoscientific Model Development, as it presents the final product obtained after certainly a lot of developments (I acknowledge the work of creating a new configuration). On the scientific side, I find the paper very descriptive, and sometimes speculative, and would therefore welcome more discussion or experiments to understand the origins of specific results (I'm giving a few suggestions below)

I appreciate the effort to develop this configuration as part of a community effort (NEMO) and believe that the results are satisfactory enough to publish this study. However, I have two main points that I would like the authors to make clearer, possibly by running one or two additional short experiments:

1. By prescribing a constant sea ice salinity of 0.001 PSU in the model, the authors essentially switch off the halo-dynamics in the sea ice model. What is the reason that led them to that choice? Did the authors conduct tests with the full interactive halo- dynamics module of LIM active, and compared the results? I know that the Baltic sea is a particularly fresh sea, hence the ice is mostly non-salty, but why not try to let the model figure it out itself?

**During the initial setup phase we tested the full halo-dynamic sea-ice option but it would crash the model near river mouths so we opted for the bulk salinity option, which yielded a stable model. We have modified the text slightly to point out this:"In our setup we neglect all internal halodynamical processes of the sea ice, as initial tests using this option yielded in an unstable model, particularly close to river mouths. Instead we use a constant bulk …"**

2. The authors diagnose the proportion of ridged ice as the ice concentration in the thickest category (p. 5, lines 16-21). The authors correctly write that this is an approximation. I'm wondering if the authors could go a bit more quantitative here. For example, by running the same simulation but deactivating the ridging/rafting scheme, they could measure the amount of thermodynamically-grown ice in the fifth category. If that amount is very small (say, < 5 %) then indeed that category in the reference run contains most of the deformed ice.

**Following the recommendation, we performed a sensitivity test (No ridging/rafting) where we ran the model without the ridging/rafting calculations, simply by omitting the call to mechanical redistribution routine (lim_itd_me). We ran the test for three years (2000-2003) and compare it to a control simulation that is identical to our 1961-2006 experiment but starts in 2000. The effect on the ice thickness distribution, of turning off mechanical deformation, is shown in Fig 1. Clearly the ridging peak centred around 3.2 m is absent in the No ridging/rafting run, as expected. However, the distributions also show a shift towards slightly thicker ice, for the No ridging/rafting run, for the lower ice classes, presumably due to the missing transfer of ice towards the thickest ice category. We further compared the ice volume in the different ice categories for the two experiments. Clearly the ice volume in the fifth ice category is much lower in the No ridging/rafting run. Integrated over the total model domain the No ridging/rafting run has 6-20% of the control run's ice volume in the fifth class. It is also seen that the ice volume is increasing throughout the season. This occurs in the Bay of Bothnia where we have the thickest thermodynamically grown ice in the Baltic Sea.**

Other comments:
 - How did the authors come to the specific values of LIM for table 1? How did they tune this configuration? Providing a bit of background, with lessons learned, would be very helpful to those authors that may want to run their own configuration (e.g. in Hudson Bay).
**We have added a little bit more text in the description column of Table 1 giving references and justification for the values we choose.**

- Given the high resolution of ~4 km, the continuum hypothesis for the sea ice medium is not necessarily true. The (elastic) viscous-plastic rheology used in LIM3 heavily rests on that assumption, though. Did the authors experience numerical instabilities? Did they check the sea ice velocity/deformation fields? What would be their advice in terms of rheology at that high resolution?
**We did not experience any numerical instabilities with the current ice settings. Inspecting ice velocity fields, and previous modelling work with other models of similar resolution, suggest that using the EVP rheology at this resolutions works fine. However, our analysis has not focused on the dynamical performance of the model, and we therefore avoid giving advice on this issue.**

- What are the ocean and sea ice model time steps?
**The ocean model time step is 360 seconds (6 minutes) and the ice model is called every 5th time step with nn_fsbc=5, so 30 minutes. We added, "The ocean model time step is 360 seconds and the ice model is called every 5th time step." to section 2.1.1.**

- Fig. 12 suggests that the simulated volume overestimates the observational reference BASIS. However, as explained in the text, BASIS is essentially a measure of level ice volume. Given the fact that the atmospheric forcing is already cold, the authors conclude that the simulated volume is definitely too low. I have two questions here: 1) Are there uncertainties available around the BASIS estimates? 2) What does the mismatch in simulated and observed volume tell about the sea ice model biases? If I understand, this is more a thermodynamic issue since the deformed ice was not considered. How would the authors do to investigate the origin of this bias? Is it possibly related to the conductivity of snow for which could be significantly different from the one used for the Arctic? See also my next comment
**There are clearly large uncertainties in the BASIS data set since the underlying data comes from many different sources. In addition, BASIS ice thickness data were originally indexed by numbers from 1–9. These numbers were assigned to thickness classes (1-2cm, 3-6cm, 7-12cm, 13-20cm, 21-30cm, 31-42cm, 43-56cm, 57-72cm and more than 73cm). Thus a lower bound for the uncertainty when it comes to ice thickness is the precision given by these classes. We have included more information and a discussion on the uncertainties in section 2.4. Since regridding the that we consider the mismatch to be within the uncertainties. As you point out any mismatch here is solely due to thermodynamics.**

- In the conclusion, the authors note that the cold bias in SST in the model is somehow contradictory with the negative bias in sea ice volume, and point towards the possible role of snow. I assume that the snow conductivity was set to its default value. However, is there a good reason to assume that snow conductivity in the Baltic Sea is the same as that of the polar regions?
**Yes, for the thermal conductivity of the snow parameter we use the default value 0.31 J/s/m/K. According to Yen (1981) the snow conductivity can be expressed as $K_s = 2.222362*(rho_s)**1.885$, where $K_s$ is the conductivity and $rho_s$ the density of snow. If we use densities for dry new snow (0.225 g cm-3) and compact/wet snow (0.450 g cm-3) we get a range $K_s=0.13–0.49$, which shows that the default value represents a value in between dry and wet snow. If snow conditions tend to be colder and drier then our value is too low. From unpublished field observations of snow density (see figure 3) it is clear that the snow density changes over the range 0.13–0.55 g cm-3 on a time scale of 6 days. It is therefore hard to justify a lower or higher value of $K_s$ than the one we use now. For future studies one could try to tune the model using different $K_s$. We have added a short comment on this in the model description section 2.1.**

Minor comments

p. 1, l. 16: "... the ice extent can reach a coverage of almost 100% during severe winters". Ice extent has usually units, which makes this sentence confusing. Consider
"ice coverage reaches 100% during severe winters" **We tried to clarify by changing to: "However, interannual fluctuations around this mean are very large and during severe winters the entire Baltic Sea can be completely ice covered (Leppäranta and Myrberg, 2009; Vihma and Haapala, 2009). "**

p. 3, l. 19: To how many vertical levels does the configuration correspond? Please specify.
**In section 2.1.1 we have now specified the number of layers:" The vertical resolution is 3 m in the upper layers, down to 60 m, and then gradually increases to 22 m at depths, with a total of 56 layers.".**
p. 4, l. 5: "include" –> "included" **Changed accordingly.**
p. 4, l. 8: "uses" –> use **Changed accordingly.**
p. 4., l. 26. I don't understand the sentence saying that the scale of most sea-ice models is ~ 1 km.

Most large-scale sea ice models currently run at ~10 to ~50 km, which is at least one order of magnitude larger than what is written. **We changed to: ",the scale of the present sea-ice model,"**
p. 5, l. 27: "show" –> shows **Changed accordingly.**
p. 7, l. 25: An extra "the" is present. **Removed.**
Fig. 4. The color bar is misleading because "100% ice" and "no ice" have essentially the same colour, i.e. white. Consider a more adapter colormap. **We have kept the colormap since it is the one recommended for ice in the http://matplotlib.org/cmocean/ package, however, we changed the background color to gray to avoid confusion with the highly ice-covered areas.**
Fig. 8a. The figure 8a seems to suggest that the sea ice model grows too much ice in the thin categories. Did the authors try to run sensitivity tests with difference ice thickness boundaries? It would be interesting to at least identify possible candidates to explain the striking differences seen in Fig. 8a. This is somehow done in the text (p. 10, first paragraph) but in a very descriptive and speculative way. **To investigate this further we ran a couple of short 5-years simulations where we changed the limits of the ice categories by changing the rn_hicemean parameter (0.25, 0.50, 0.75, 1.50 m) and the number of categories (from 5 to 10). This is presented in figure 4. It is seen that the only case when the simulated ice thickness distribution changes significantly is for the case with 5 categories and rn_hicemean=0.25. Here all the ice categories are basically only resolving the thinner ice, which leads to more ice in the range 1–3 m. However, this seems to be an artefact as the case with rn_hicemean=0.5 and 10 categories also resolve the thinner range with a high resolution but lacks the peak around 1.0 m. To further investigate this, experiments with different transfer functions could perhaps be a way to improve the distribution.**

**Response to reviewer 2**

Overall this was an interesting description of an ice-ocean modelling system for the Baltic Sea. I do have a few concerns, but feel that the manuscript may be acceptable after some relatively minor revision. There were no scientific hypotheses in this paper, so I am simply evaluating the technical detail aspects.

1. My first concern is about the model boundary conditions. I understand that this domain is forced from a larger North Atlantic domain and that the ocean exchange is not a key part of the Baltic Sea ocean temperature and salinity structure. However, what happens with fresh water from the atmosphere and land? What impact does the river discharge have on the sea surface temperature and salinity of the Baltic? How is this specified in the modelling system?

**The large net freshwater inputs from the rivers and the atmosphere are important drivers of the estuarine circulation of the Baltic Sea. This leads to a strong low saline outflow from the Baltic Sea towards the North Sea. The freshwater sources shape the Baltic Sea hydrography an creates a permanent halocline, with small seasonal variations. Since the system is brackish the salinity stratification limits the upper convections in regions where the temperature of maximum density is larger than the freezing temperature. The model has a nonlinear free surface that handles the freshwater input as a mass flux. Having an open boundary close to the region of interest is of course not optimal, but previous modelling efforts e.g. Meier (2011) have shown that this approach works well for the Baltic Sea. In addition, the simulated and observed hydrography are very similar giving merit to this approach. However, for the Kattegat region the approach has some problems, as we see in our ice analysis.**

2. Despite the SST biases as described in the manuscript, the model does a pretty nice job at simulating the sea ice concentration, thickness, and extent. I wonder about the snow on the sea ice in this region? I know the authors present a climatological seasonal cycle of snow from the model, but I would like to see more discussion here. Can the authors add the observed snow depth information to Figure 15? Are there no other sites or model information to get snow depth? Maybe accumulating snowfall from the forcing? As the authors know, the albedo of the snow is critical to the seasonal evolution of the sea ice. Albedo formulation? Parameter settings?

**We have now gathered snow observations from two stations and added this to section 3.1. The model uses the Shine and Hendersson-Sellers (1985) albedo formulation that separates between melting and freezing snow and ice. We use the standard settings of this implementation in LIM3, see the user guide http://www.climate.be/users/lecomte/LIM3_users_guide_2012.pdf .**

3. One thing I don't know about the Baltic is whether the daily maximum sea ice extent always occurs in the same month? It is sometime in February or March. It is interesting in Figure 7 that the model maximum extent is in March, while the observations indicate February. There should be more discussion of this.

**No this can vary from year to year but mostly commonly it occurs in February or March, but sometimes also in January. We have followed your recommendation and added a paragraph in section 3.2 where we discuss this. From the analysis we see an average offset of 9 days which has to be considered good since the ice charts are produced 1-2 per week.**

4. From Figure 8, it looks like the model has too much thin ice (less than 1m) and not enough thick ice. How does this vary from year to year? I wonder if there is a relationship to snow here. Is there a freeboard parameterization in the model? Are you getting to much or too little snow-ice formation?

**The general features of Figure 8 are seen throughout the simulation and indicate a systematic problem unfortunately. The model has capabilities to estimate negative freeboard and snow-ice formation which is included in the total ice production. We do not have saved information of snow-ice formation for the entire run. However, from a 5 year test run we see that the snow-ice formation, average over the larger basins, contributes with 4-6% of the total ice growth, depending on the location. The main contributor to ice growth is bottom growth, followed by snow-ice and lateral ice growth. Unfortunately, we do not have observations to compare these simulated values of snow-ice growth with.**

5. Finally, there are a lot of figures (sixteen) given the amount of text in the manuscript. A number of the figures are very qualitative, just comparing contour plots by eye. Could some of these be condensed into more quantitative information, maybe spatial correlations or differences between the model and observations? Just a thought.

**Three figures have been removed, the FDD comparison, the snow thickness seasonal cycle and the January and February level ice thickness maps. However, we also added two more figures so the total is now 15. We agree that there are many figures and that the analysis is sometimes too qualitative rather than quantitative. However, we tried to be more quantitative in the text and think that these figures best illustrate our analysis**

**Response to reviewer 3**

The paper is based on good and original material, and deserved to be published. Writing is generally good. I found the paper material quite interesting.

Yet I believe it should be improved, following two categories of objections.

1) The conclusions could be more general and interesting.

2) The analysis of results could be more acute and precise

Besides, some sections (3.1, 3.4) are not fully clear and could be sharpened.

I think the manuscript can easily be improved, and I hope my comments will help.

*** General comment #1. The conclusions could be more generic.

I would somehow use the results of the paper paper to question the capabilities of regional modelling ice systems forced by atmospheric reanalysis, considering Baltic Sea ice as a successful example.

The following core of conclusions could be the base of the abstract/

Conclusion 1: The NEMO-Nordic ice modelling system is appropriate to get the mean extent, volume, and geographical distributions of ice concentration and thickness in the Baltic Sea, which all seem rather precisely captured (within 10% of obs?). The ice melts early, which is attributed to XXX.

Conclusion 2: Extreme years, in particular severe winters, are more difficult to simulate.

Conclusion 3: The subgrid scale ice thickness distribution seems challenging.

**We have used your conclusions 1 and 3. After regridding the observational data set conclusion 2 does not hold anymore. And the limits are also somewhat arbitrary. The correlation and standard deviation are well-captured, but the trend is lower due to problems in Kattegat.**

*** General comment #2. The analysis of results could be more acute and precise

- The SST bias should be analyzed with respect to ice presence or absence. In presence of ice, the SST must be very very close to the freezing point, hence any SST bias has to be attributed to SSS. The warm bias in April is very likely due the early ice retreat, which was barely mentioned in the text. **Yes we agree. We have toned down the satellite SST analysis in the ice cover areas since it is uncertain what exactly what the data show in these areas. Using the observed ice cover we have stippled these areas to make it clearer.**

- Why ice melts too early is not clearly attributed. There are admittedly bits and pieces, but the analysis could be more systematic (air temperature, snow depth, incident solar radiation, surface albedo). The ice thickness bias is neither clearly quantified. **We have included 2 m air temperature observations to further investigate this, but since we lack observational estimates on other important heat flux components we fail attributed what causes the observed simulated anomalies.**

- The ice thickness distribution analysis would be more conclusive if (i) the model ice categories were used for both observations and model, (ii) the exact same time and locations were used to construct the pdf. At this stage, the model looks really bad, but this could be because different ice categories are used. **We have binned the data the same way as Löptien *et al* (2013) to highlight the differences between the two models. Binning both data sets using the models ice category bounds essentially show the same features, but with less detail, see Figure 5 below.**

- The comparison of snow at two stations is not enough to conclude, as blowing snow effects can be locally dominant. As the present analysis is not meaningful, you could remove it and just mention it as being inconclusive. Snow depth could be critical and I'm surprised there are not more snow depth observations to compare with, in such a well studied coastal sea. If available, a more systematic snow depth analysis would be a real plus. **We have now changed the snow depth comparison to include observations from two stations. These were the only on-ice based stations we could find. Admittedly, comparing only two stations is not enough for robust conclusions, but both these stations have quite long time series and indicate that there seems to be a systematic underestimation of the snow the in the forcing. Samuelsson et al also found this for some land-based stations around the Bothnian Gulf. We also tried to use nearby land-based**

stations, but here the comparison is even more questionable when we use ocean grid points, so we avoid presenting them.

- I think the FDD-model analysis is confusing and adds unnecessary complexity to the paper. I would recommend to compare the winter air temperature bias instead, that would be simpler and actually equivalent. **We agree, the FDD comparison was somewhat "artificial" and has now been removed. We have also add a short air temperature bias comparison in section 3.1.**

- The effect of fast ice parameterization is claimed to improve the results, but this is not supported by material. Therefore, I would recommend to be more explicit. **The main purpose of the simple fast ice parameterization is to displace the thickest ice off the coast. To illustrate this better we have added an inset with a zoom over the Bothnian Bay for the simulated March distribution, and tried to be more explicit on this in the text.**

- The analysis of extreme years is quite interesting. It should be stated or shown whether the analysis applies to all extreme ice years. Do all severe years follow a similar ice thickness pattern? Are all mild years realistically captured. In addition, whether forcing or model are responsible should be at least discussed. **Since regridding the data the statistics shifted somewhat. We have now commented a little bit more on this. Identified problem areas and provided an animation in supplemental materials showing the year to year changes in MBI distribution.**

- Be more quantitative in the text in general (give numbers instead of "reasonable" or "quite good") **Ok, we have tried to give more quantitative measures and toned down some of the qualitative assessments of the data.**
- Revise your conclusions once analysis has been sharpened. **Ok.**

*** Detailed comments
- Introduction can be sharper and a better selection of the required elements could be done. **We have added a reference and a few small text changes, but since no explicit suggestions where maid we like to keep it as it is.**
- p4, l.5 "include"D" **Changed accordingly.**
- p4. l.17 give reference, because 0.17m is not the common value in LIM3. In practice, you removed rafting from the model. Why did you do that ? **We changed the crossover thickness based on analytical work by Parmerter. We changed the text slightly to point this out: "likewise we lowered the crossover thickness for when sea ice ridge instead of raft from 0.75 m to 0.07 m, with a more sharp transition. This value is a Baltic Sea adaption of the analytical modeling work by Parmerter (1975), who suggest 0.17 m for Arctic conditions. "**

- p.4 your fast ice parameterization is grid-size dependent because your criterion is based on cubic meter. Why did not you use a volume of ice per grid cell area criterion ? **We based this on the parameter "zvmin = 1.0e-03_wp ! ice volume below which ice velocity equals ocean velocity ", this can clearly be improved.**
- p.5 your mean thickness is now referred to as volume per unit area (Notz et al, TC 2016). **Ok.**
- p.5 line 14, why do you use 5 and not 4 in the denominator to compute your level ice thickness ? **Typo, we have now changed to 4.**
- p.6 line 20. Could you describe in one sentence what is the observation based for ice charts ? satellite ? visual ? how was ice thickness quantified ? **The ice charts used for the BASIS ice product were derived from direct ice measurements and estimates from voluntarily observing ships, coast guards, ice breakers, light houses and harbour authorities. Additional information came from over-flights by FMI, SMHI and the Swedish Air Force. We have included more information and a discussion on the uncertainties in section 2.4.**

- section 3.1 should probably be revised, I found it hard to follow (see general comment as well). **We have changed section 3.1 now so that it includes analysis of air temperature, sst and snow thickness biases.**

- p. 9, l. 9 "variability" -> "interannual variability". "quite well" -> be more quantitative. Are you sure the units of STD are correct ? **Ok. Yes the units missed a 10\*\*3.**

- p. 9 l. 13. What about the trend if you exclude the first 15 years ? May be use ice area to see if that is robust ? **Since we regridded the data Figure 8 has now changed a bit. Calculating the trend for 1977-2006 yields -32 (-27) 10\*\*3 km\*\*2, which can be compared -23 (-10) for the full period, for NEMO Nordic (BASIS/IceMap). We added a short comment on this.**

- p.10 I can't reconcile the file that volume could be overestimated with the fact that the ice melts too early. **Now that we have regridded the data and compare it to the "level" ice volume it agrees better. However, early melt is still seen there.**

- p. 11, l 16 "consistS" **Changed accordingly.**

- section 3.4. If you keep this analysis, which I don't especially recommend, you may want to explain why you use different sites for FDD and for snow depth. I finally figured why, but it would have been good if you had said it right away. **We followed your advice and have now removed the FDD section**.

- acknowledgements: acknowledge NEMO developers :-) **Done!**

**Figures**

[Figure]

**Figure 1.** Mean March ice thickness distribution calculated over the Bothnian Bay for the period 2000–2003.

[Figure]

**Figure 2.** Mean ice volume integrated over the five different ice categories for January–April 2000–2003. The first columns for each month represent the control run and the second columns the test run with sea-ice deformation turned off.

[Figure]

**Figure 3.** Unpublished snow density measurements during the SEAICE09 cruise (5[th]-11[th] of March 2009) in the Bothnian Bay.

[Figure]

**Figure 4.** Simulated ice thickness distributions compared to the EM-data distribution for a control case (top), rn_hicemean=0.25 m and 5 categories (middle left), rn_hicemean=0.75 m and 5 categories (middle right), rn_hicemean=0.50 m and 10 categories (bottom left) and rn_hicemean=1.50 m and 10 categories (bottom right). The distributions are based on data for the EM-data dates in 2003, 2004 and 2005.

[Figure]

**Figure 4.** Simulated and observed ice thickness distributions, same as Figure 8 in the manuscript but binned with ice category bounds as limits.

**References**
Yen, Y.-C. (1981) Review of Thermal Properties of Snow, Ice and Sea Ice (CRREL Report 81/10, pp. 1-27). U.S. Army Cold Regions Research and Engineering Laboratory, Hanover, NH.

[revised manuscript text omitted]

---

## Author Response (AR2)

RESPONSE TO TOPICAL EDITOR

Below we have answered the comments raised by the topical editor, our answers highlighted by the bold black text. After the answers a LaTeX-diff, showing the new text additions as blue text, is included.

Best regards,

Per Pemberton

On behalf of the authors

Referee 1, "2. The authors diagnose the proportion …"

The response here is fine, but I feel that there is a case for mentioning the failure of this test, at least in passing. Sometimes model improvements turn out not to be improvements, and noting this could save other workers wasted effort.

**We have added two new sentences on page 5 lines 23-26: "In an effort to assess the precision of this approximation we ran a 5 year test where we turned of all mechanical deformation of ice, and then compared it to a control simulation for the same period to get an estimate of thermodynamically grown ice in the fifth class. This resulted in ice volumes (in category five) of 6--20\% in the case with no deformation, however, the ice in the lower ice classes was also strongly impacted due to the missing transfer of ice, and the test was deemed too artificial to assess the precision of the approximation."**

Referee 1, "p. 3, l. 19: To how many vertical levels …"

I would suggest deleting the comma after "upper layers", and switching "depths" -> "depth" (i.e. "... to 22 m at depth, with ...")

**Changed accordingly.**

Referee 3, "General comment #2 …"

A figure change is mentioned in your response, but the referee mentions aspects that do not appear to be addressed (or are not stated as being addressed). For instance, the SSS bias and early ice retreat. Can you please clarify whether these have been addressed?

**Reviewer 3 pointed to the fact that in ice-covered areas the SST should always be close to freezing, and hence the bias must either be in the SSS (or most probably in the atmospheric forcing). To avoid these problem areas we masked them out using the observed ice cover. In addition, there are very few observations of SSS in ice covered areas, and to produce reliable statistic would have been a challenged. Therefore, we avoid these areas. So approach to address this issue was simply to avoid them. This was perhaps not very clear in our answer.**

Referee 3, "Why ice melts too early is not clearly attributed …"

Again, a figure has been added, but it is unclear what other changes have been included. The latter point about observational estimates could be fashioned into a limitation or caveat of the study and added to the discussion to prompt future work.

**Since we lack information on the heat fluxes we can not fully disentangle the causes of the model offsets. In the revised manuscript we point to this in several places: page 8 line 6, page 10 line 23, page 12 line 21, page 14, line 25, and page 17 line 3. We have now also added two sentences in the conclusion/summary section on this limitation, see answer below.**

Referee 3, "The comparison of snow at two stations …"

Have you remarked on the lack of availability of data in the text on this point? Again, it may be a useful discussion point to drive future work.

**We caution the reader on page 9 line 10 on this issue. We have now also added two sentences in the conclusions/summuary section:" The lack of reliable long-term and spatial representative observational data in ice covered regions limit our study to fully explore the model performance.**

**Here particularly more snow and ice thickness data, as well as, radiation data would greatly improve future sea ice model assessments in the Baltic Sea." This is included on page 17 lines 19-21.**
Referee 3, "Revise your conclusions once analysis has been sharpened."

What has been done here? Your response, "Ok", is fairly non-specific. Are you referring to General Comment #1 of this referee?
**This answer is agreeably very non-specific. To clarify, we revised our conclusions following reviewer 3´s suggestions 1 and 3, as we explain in that answer.**
* * *
Code availability

While your current text notes the availability and revision number of the underlying NEMO code that your work utilises, it does not document the modifications that you have made to perform the work in this manuscript to the same standard. For the latter, it currently reports the following:

"Our Baltic Sea adopted code, input data, analysis scripts and data used to produce the figures in this study can be made available upon request by the corresponding author."

The nature of the "adopted code" ("adapted code"?), is unclear here, no reference is made to whether it is under version control, or, if so, which revision is used in the work here. If the adapted code consists of modified versions of existing NEMO routines, perhaps these could be made available as a supplement for this paper. The same would be possible for wholly new code blocks. If it is not possible to make the code (and the parameter sets or other input data) available in this way, it is important to note why this is. For reference, GMD's guidance on code availability runs as follows:

"All papers must include a section, at the end of the paper, entitled "Code availability". Here, either instructions for obtaining the code, or the reasons why the code is not available should be clearly stated. It is preferred for the code to be uploaded as a supplement or to be made available at a data repository with an associated DOI (digital object identifier) for the exact model version described in the paper. Alternatively, for established models, there may be an existing means of accessing the code through a particular system. In this case, there must exist a means of permanently accessing the precise model version described in the paper. In some cases, authors may prefer to put models on their own website, or to act as a point of contact for obtaining the code. Given the impermanence of websites and email addresses, this is not encouraged, and authors should consider improving the availability with a more permanent arrangement. After the paper is accepted the model archive should be updated to include a link to the GMD paper."

**Our code contains just a few really minor changes which are stated in the beginning of the code availability section. We have it under revision control and can provide a revision number. The repository is available online but a user account is required to download the code. If we provide the link to the repository and an email to me or the admin person, it is perhaps not an optimal solution, but more convenient for us, would that work? If not, then we will provide the relevant subroutines or code blocks as supplemental material.**

[revised manuscript text omitted]

---

## Author Response (AR3)

Below we have answered the comments raised by the topical editor, our answers highlighted by the bold black text. After the answers a LaTeX-diff, showing the new text additions as blue text, is included.
Best regards,
Per Pemberton
On behalf of the authors

I think that the only sticking point for me is the code availability section. You replied previously as …

"Our code contains just a few really minor changes which are stated in the beginning of the code availability section. We have it under revision control and can provide a revision number. The repository is available online but a user account is required to download the code. If we provide the link to the repository and an email to me or the admin person, it is perhaps not an optimal solution, but more convenient for us, would that work? If not, then we will provide the relevant subroutines or code blocks as supplemental material."

Firstly, you overlooked my point about "adopted" (i.e. code you've taken from elsewhere) vs. "adapted" (i.e. code from here you've modified). That ambiguity still remains in the main text. Can you clear it up please? **Sorry for this. I have now changed the "Baltic Sea adopted code" to NEMO-Nordic code. We have not really changed the standard NEMO code, rather just extended it to include some new features, which we need for the Baltic Sea. Hope this clears things up.**

Regarding your suggestions around the modified code, reporting the repository address, the revision number and supplying a contact point to get an account should be enough for this. The latter would probably best be an administrative contact point rather than an individual, just in case the individual moves to a new job, etc. The information around the repository and revision should be intelligible and sufficient for the contact point to work out what's being requested some time down the line when this paper isn't fresh in everyone's minds. Refer back to the guidance I included last time for more pointers. **I have now included the repository and revision number used in this work. However, we do not have a system where we can provide an administrative contact point, so the contact point has to be me I am afraid. The Subversion repository is mainly for internal use, and for our collaborators at the moment. The code availability paragraph now reads:**
**"**
*NEMO-Nordic builds on the standard NEMO code (nemo_v3_6_STABLE, revision 5628) with only minor changes including: the fast ice parametrization and a spatial varying background viscosity/diffusivity that could be read in from file. The standard NEMO code can be downloaded from the NEMO web site (http://www.nemo-ocean.eu/). The nemo_v3_6_STABLE version is available from the following link:*
*http://forge.ipsl.jussieu.fr/nemo/svn/branches/2015/nemo_v3_6_STABLE. The new code blocks that are introduced (relative to the standard NEMO code nemo_v3_6_STABLE, revision 5628) into our NEMO-Nordic code are included as supplemental material. The full NEMO-Nordic code is in a Subversion revision control system repository, available under*
*http://54.73.141.37/subversion/repository/source_code/trunk/NEMOGCM. However, a user account is needed to gain full access. This work used the revision 339 of NEMO-Nordic code. Access to the NEMO-Nordic code and all input data, analysis scripts and data used to produce the figures in this study can be made available upon request by the corresponding author.*
**"**
 **Note that we have also removed the statement "***and changes that allow for a simulation start time other than midnight***" from the Code availability section, since it doesn't apply to these simulations. It is only required when the model is run in forecast mode and there is a need to start the model several times a day.**

And, to be honest, unless it's a big pain, I would be inclined to add the modified code blocks as supplementary material in addition. **This is now included, see additional supplemental material.**

[revised manuscript text omitted]